# Development of an Ergonomic Additively Manufactured Modular Saddle for Rehabilitation Cycling

**DOI:** 10.3390/ma18225242

**Published:** 2025-11-19

**Authors:** Alberto Iglesias Calcedo, Chiara Bregoli, Valentina Abbate, Marta Mondellini, Jacopo Fiocchi, Gennaro Rollo, Cristina De Capitani, Marino Lavorgna, Marco Sacco, Andrea Sorrentino, Ausonio Tuissi, Carlo Alberto Biffi, Alfredo Ronca

**Affiliations:** 1Institute of Polymers, Composites and Biomaterials (IPCB), National Research Council (CNR), Via Gaetano Previati, 1/E, 23900 Lecco, Italy; albertoiglesiascalcedo@cnr.it (A.I.C.); valentina.abbate@cnr.it (V.A.); cristina.decapitani@cnr.it (C.D.C.); marino.lavorgna@cnr.it (M.L.); andrea.sorrentino@cnr.it (A.S.); 2Institute of Condensed Matter Chemistry and Technologies for Energy (ICMATE), National Research Council (CNR), Via Gaetano Previati, 1/E, 23900 Lecco, Italy; chiara.bregoli@icmate.cnr.it (C.B.); jacopo.fiocchi@cnr.it (J.F.); ausonio.tuissi@cnr.it (A.T.); carloalberto.biffi@cnr.it (C.A.B.); 3Institute of Intelligent Industrial Technologies and Systems for Advanced Manufacturing (STIIMA), National Research Council (CNR), Via Gaetano Previati, 1/E, 23900 Lecco, Italy; marta.mondellini@cnr.it (M.M.); marco.sacco@stiima.cnr.it (M.S.); 4Institute of Polymers, Composites and Biomaterials (IPCB), National Research Council (CNR), P.le E Fermi 1, 80055 Portici, Italy; gennaro.rollo@cnr.it

**Keywords:** additive manufacturing, lattice structures, rehabilitation devices, cycle ergometers

## Abstract

This work reports the design, fabrication, and validation of a modular ergonomic saddle for rehabilitation cycling, developed through a combined additive manufacturing approach. The saddle consists of a metallic support produced by Laser Powder Bed Fusion (LPBF) in AISI 316L stainless steel and a polymeric ergonomic covering fabricated via Selective Laser Sintering (SLS) using thermoplastic polyurethane (TPU). A preliminary material screening between TPU and polypropylene (PP) was conducted, with TPU selected for its superior elastic response, energy dissipation, and more favourable SLS processability, as confirmed by thermal analyses. A series of gyroid lattice configurations with varying cell sizes and wall thicknesses were designed and mechanically tested. Cyclic testing under both stress- and displacement-controlled conditions demonstrated that the configuration with 8 mm cell size and 0.3 mm wall thickness provided the best balance between compliance and stability, showing minimal permanent deformation after 10,000 cycles and stable force response under repeated displacements. Finite Element Method (FEM) simulations, parameterized using experimentally derived elastic and density data, correlated well with the mechanical results, correlated with the mechanical results, supporting comparative stiffness evaluation. Moreover, a cost model focused on the customizable TPU component confirmed the economic viability of the modular approach, where the metallic base remains a reusable standard. Finally, the modular saddle was fabricated and successfully mounted on a cycle ergometer, demonstrating functional feasibility.

## 1. Introduction

Additive manufacturing (AM), commonly known as 3D printing, is widely regarded as one of the most transformative manufacturing technologies of the 21st century. Its growing adoption across industries results from its material efficiency, design flexibility, geometric freedom, and shortened production cycles [1,2]. Among the main AM techniques used in the industry are vat photopolymerization (VPP), powder bed fusion (PBF), and material extrusion (MEX) [3]. According to the Wohlers Report 2023, polymer-based PBF (e.g., SLS, MJF, SAF) dominates the industrial market, whereas MEX remains prevalent among consumer applications. LPBF and SLS are widely used for processing metals and thermoplastic polymers, respectively. They have become the techniques of choice in applications such as customized prosthetic sockets and orthotic supports [4,5,6,7,8]. These methods also enable the implementation of generative design, topology optimization, and bio-inspired architectures [1,9]. Such architectures, typically composed of prismatic or polyhedral networks, can be precisely tuned to achieve the desired mechanical performance while minimizing weight and material usage [10,11]. Cellular materials take inspiration from biological systems, where natural structures such as sponges, corals, wood, and trabecular bone exhibit complex networks of interconnected struts and plates [12]. Gibson and Ashby’s seminal classification distinguishes between open-cell foams, closed-cell foams, honeycombs, and lattice structures, the latter exhibiting distinct topological characteristics and property profiles [13]. In this scenario, lattice structures such as Voronoi or gyroid geometries can be strategically designed to optimize pressure distribution, enhance comfort, and provide mechanical support [4,14,15,16,17].

Cycling-based rehabilitation is widely recognized as an effective method for restoring lower limb function and cardiorespiratory fitness after injury or surgery, due to its low-impact and controlled motion characteristics [11,18,19]. However, most commercially available saddles are optimized for performance rather than for therapeutic comfort or prolonged stationary use. In rehabilitation settings, inappropriate saddle geometry or stiffness can cause discomfort, poor posture, or uneven load distribution, potentially compromising patient compliance and biomechanical outcomes [20]. Despite the advances in rehabilitation technology, customized and ergonomically adaptive saddles for rehabilitation cycling remain extremely limited. AM provides a unique opportunity to fill this gap through patient-specific geometries and localized mechanical tuning, improving both comfort and mechanical safety during training.

In response to this gap, there is growing interest in developing innovative saddle solutions that combine ergonomic design with advanced manufacturing technologies. Although several companies have applied AM to develop high-performance cycling saddles, such as Stratasys, which offers trabecular-structured saddles for competitive cyclists, these designs focus primarily on performance rather than therapeutic benefit [21,22,23]. For example, the AM saddle prioritizes impact resistance and lightweight design, while others use lattice structures to increase comfort without modularity. However, rehabilitation cycling remains underserved. Saddles for rehabilitation bicycles are typically standard models with soft padding and minimal adjustability. These conventional designs do not meet the diverse postural and anatomical needs of patients, particularly those undergoing daily therapy. Individual support requirements, such as asymmetric padding or targeted pressure relief, are often overlooked.

To address these limitations, this study presents the design and development of an additively manufactured modular saddle optimized for rehabilitation cycling. Two powder bed fusion technologies, LPBF and SLS, were used to fabricate the saddle’s metallic and polymeric components, respectively. LPBF was used to produce the metallic support structure in stainless steel (AISI 316L), providing high strength and geometric complexity. The selected lattice structure offers a good balance between strength, lightness, and additional functionalities such as damping behaviour. This feature is particularly promising for sport-oriented components, which are exposed to dynamic loads where comfort is essential [24,25,26]. In parallel, SLS was used to fabricate the ergonomic polymeric covering. To determine the most suitable material, two thermoplastic polymers, TPU and PP, were evaluated. Both materials were 3D printed into gyroid lattice structures and tested under uniaxial tensile, compression, and fatigue loading. Differential scanning calorimetry (DSC) and thermogravimetric analysis (TGA) were also performed to optimize SLS printing parameters. These analyses guided the final material selection and processing strategy. The proposed design aims to improve user comfort, accommodate different body types, and reduce the risk of pelvic floor discomfort, muscle fatigue, and nerve compression during prolonged rehabilitation sessions. This research introduces an innovative approach to saddle design by exploring lattice architectures, material combinations, and design strategies to identify the most effective configuration for therapeutic use.

## 2. Materials and Methods

### 2.1. Design Process

Before initiating the production phase, it was essential to define the core design principles of the modular saddle system. In particular, the saddle consists of two main components: a metallic structural support and a polymeric ergonomic covering. Both were fabricated using AM technologies but required distinct design strategies, materials, and process parameters to allow for modularity and customization, a single metallic support combined with a customizable polymeric covering offering a patient-oriented modular system as shown in Figure 1a. These dual-design paths necessitated a combined AM approach where each component was developed and manufactured following a dedicated workflow, described in the following subsections.

### 2.2. Metallic Support: Design and Fabrication

The external geometry of the metallic saddle support was inspired by standard rehabilitation bicycle saddles, and the bulk shape was initially modelled using Autodesk Inventor. The model was then exported to 3-matic (Materialise, Leuven) for the application of a Voronoi tessellation, chosen for its isotropic structure, mechanical robustness, and smooth, edge-free morphology. The Voronoi lattice was designed with the following parameters: a strut thickness of 4 mm and cell spacing of 10 mm.

The final model included the following: two slits for mounting the polymeric coverings and three holes for attachment to the rehabilitation bike frame. The completed STL model was then prepared for LPBF printing (Figure 1b).

An LPBF system (mod. AM400 Renishaw) equipped with a pulsed wave laser was used to produce AISI 316L samples. Optimized processing parameters were used (see Table 1); a density equal to 99.6% of relative density was achieved. After printing, all the samples were subjected to heat treatment at 800 °C for 1 h followed by air cooling; these conditions were chosen in previous work. A post-printing sandblasting surface finishing was applied for surface finishing. Threading was then performed on the mounting holes to allow for mechanical fastening.

After LPBF fabrication, the stainless steel base was sandblasted to remove loosely bonded particles and ensure dimensional compatibility with the polymeric components. No surface roughness measurements were performed, as the metallic element acts solely as a structural support and is not intended for direct contact or functional load transfer.

### 2.3. Polymeric Covering Design and Material Selection

Bicycle saddle width plays a crucial role in cycling comfort and performance. Research indicates that wider saddles generally provide better pressure distribution and comfort for cyclists, particularly for women [27]. A study on cyclists found that wide saddles resulted in lower maximum and average pressure on the posterior ischium compared to narrow and moderate saddles [27,28].

The optimal saddle width appears to be related to an individual’s anatomy. Lin et al. suggested that a saddle width that is 1 cm longer than the cyclist’s ischial tuberosity width can effectively improve saddle pressure distribution and comfort [27]. Other literature results study found significant correlations between inter-individual differences in ischial tuberosity width and the posterior centre of pressure location on the saddle [28,29,30].

There is no single “average” saddle width that suits all patients, but research suggests that saddles should be sufficiently wide to support both ischial tuberosities. This is best achieved when the saddle is at least two times wider than the bi-ischial width of the cyclist [29]. However, individual factors such as gender, anatomy, and riding position should be considered when selecting an appropriate saddle width to optimize comfort and reduce the risk of injury. The ergonomic padding was designed to ensure adaptability to different users, particularly considering the anatomical variations between male and female pelvic structures. According to Selle Royal, average saddle widths are approximately 125–135 mm for men and 145–150 mm for women. Pelvic width classifications are as follows: narrow (<110 mm), medium (110–130 mm), and wide (>130 mm). The smallest pelvic width was adopted as the base reference, enabling modularity and user-specific customization.

The modular design was developed using Autodesk Inventor and incorporates advanced lattice geometries optimized for stress distribution and comfort. The selected lattice geometry was the Gyroid structure for its continuous surface and mechanical resilience as reported in the literature [31,32,33].

The saddle geometry model was imported to Autodesk fusion, where the gyroid volumetric lattice was applied by varying cell dimension and lattice thickness as shown in Figure 2. Briefly, a variety of sheet gyroid-based Triply Periodic Minimal Surface (TPMS) geometries have been designed to assess how key structural parameters affect the mechanical performance of 3D-printed samples. Sheet gyroids were chosen because they are well suited for 3D printing and allow for a straightforward adjustment of mechanical response by modifying the unit cell size and wall thickness [34]. The geometries are derived from the theoretical gyroid surface, defined by Equation (1).sin(x)cos(y) + sin(y)cos(z) + sin(z)cos(x) = 0(1)
where x, y, and z are Cartesian coordinates ranging from 0 to 2π. This equation describes a minimal surface that forms a single unit cell. To generate the printable solid sheet gyroid, the surface is scaled to the desired unit cell length, repeated to achieve the required number of cells in each direction, and thickened to the specified wall thickness. Gyroid lattices were generated using the TPMS function implemented in Autodesk Fusion 360 version 2604.1.25 (Autodesk^®^, San Rafael, CA, USA). The software allows for a direct creation of TPMS-based solids from the implicit Gyroid equation by defining a parametric wall thickness. Two configurations were designed with nominal wall thicknesses of 0.2 mm and 0.3 mm, respectively. This CAD-based approach provides full control and repeatability of the lattice geometry for subsequent AM process.

The study explored unit cell side lengths of 6, 8, and 10 mm, combined with wall thicknesses of 0.2 and 0.3 mm, resulting in six distinct structural configurations. Each sample was composed of enough repeated cells to form a final cylindrical specimen size of 40 × 20 mm (Ø × h). The theoretical volumetric lattice parameters were monitored to achieve two different porosities of 79.63 ± 0.05% and 69.59 ± 0.18% for 0.2 and 0.3 wall thickness lattice structure, respectively, as reported in Table 2. The real porosity of the lattice sample was determined using a geometrical–gravimetric method. The external dimensions (Ø and h) of each cylindrical specimen were measured with a digital gauge, and the total volume (V) was calculated accordingly. The apparent density (ρ_app_) was obtained by dividing the measured mass by the total volume, while the measured bulk density of the TPU material (ρ_bulk_ = 0.89 ± 0.02 g/cm^3^) was used as reference. The porosity (P) was then calculated following Equation (2):(2)$$P\left(\%\right)=\left(1-\frac{\rho_{app}}{\rho_{bulk}}\right)\cdot 100$$

Measurements were performed in triplicate for each lattice geometry, and mean values ± SD are reported in Table 2. To simplify referencing throughout the manuscript, a concise nomenclature was adopted for the gyroid lattice specimens. Each sample name is coded as GX_Y, where G stands for “Gyroid”, X represents the unit cell size in mm, and Y indicates the wall thickness in mm. For example, the specimen with a cell size of 6 mm and a wall thickness of 0.2 mm is abbreviated as G6_0.2. This nomenclature will be used consistently in the subsequent sections for clarity and brevity.

In the preliminary design phase, a comparative material selection was conducted to identify the most suitable polymer for the ergonomic covering. Two polymeric-based powders, specifically a TPU powder (LUVOSINT X92A-1) purchased from Luvocom (Lehmann & Voss, Hamburg, Germany, 2016) and a PP powder purchased from ADVANC3D Materials^®^ GmbH (Hamburg, Germany) under the brand name AdSint PP Flex’ (PP). Both materials were selected based on their widespread use in AM and their mechanical behaviour under cyclic loading, which is critical for rehabilitation cycling applications.

To assess their suitability, both TPU and PP were employed in the fabrication of identical gyroid lattice structures, enabling a direct comparison under consistent geometric and processing conditions. The samples were subjected to a preliminary thermal and mechanical campaign, including Differential Scanning Calorimetry (DSC), thermogravimetry (TGA)-determined thermal transitions, as well as a compression test in accordance with ASTM standards [35,36].

TPU, owing to its elastomeric nature, exhibited superior elasticity, energy dissipation capacity, and dimensional recovery under cyclic deformation. In contrast, PP, although characterized by a lower density and reasonable fatigue life, displayed a comparatively higher stiffness and reduced elongation at break. These material-specific behaviours were critically evaluated in relation to the lattice geometry’s load distribution and damping functions—key aspects for ensuring user comfort and long-term structural integrity.

### 2.4. Preliminary Thermal and Mechanical Analysis

The decomposition temperatures of TPU and PP powders were identified using TGA2 Star system equipment (Mettler Toledo, Columbus, OH, USA) A small amount of powders (15 ± 0.01% mg) was heated from room temperature to 800 ± 1 °C under air environment according to the ASTM standard E1131 [37]. The degradation onset temperature (DOT) was defined as the temperature at which 1 wt.% loss occurred. The sample purge rate was 50 mL/min under air, and the balance purge rate was 20 mL/min under the nitrogen flow of 20 psi.

To establish the process temperature during the SLS 3D printing process of a powder, the melting point onset temperature should be known. To identify the melting onset temperature of TPU and PP powders, DSC tests were performed using a DSC3 Star System (Mettler Toledo, Columbus, OH, USA). TPU and PP powders (5.00 ± 0.01% mg) were encapsulated in an aluminum crucible pan (40 µL) and loaded into DSC. Initially, the temperature was equilibrated at −50 °C and then ramped at a rate of 10 °C/min to 250 °C holding isothermally for 2 min, cooling down to −50 °C at 10 °C/min, and eventually holding isothermally for 2 min. To further support the material selection process for the ergonomic lattice covering, a preliminary mechanical characterization was performed on both TPU and PP. Samples with identical gyroid geometry (G6_0.2) were fabricated and subjected to uniaxial compression tests, following the ISO ASTM D695-23 [36], to evaluate their mechanical response under uniaxial compressive loading. A strain of 15% was applied to the specimens, and the resulting stress–strain behaviour was analyzed and reported in Figure 3c.

### 2.5. Three-Dimensional Printing of the Saddle

The polymeric components were realized using a Sharebot SnowWhite SLS (Sharebot S.r.l., Nibionno, Italy). The machine uses a 200 μm CO_2_ laser (P = 14 W − λ = 10.6 μm) to sinter the polymeric powder with the possibility to allows for either chamber or bed temperature control. The printing process was conducted in air atmosphere using the bed temperature control to achieve improved dimensional accuracy and surface definition of the printed parts.

Preheating was achieved via halogen lamps positioned above the build plate. Key process parameters, such as the powder bed temperature, were optimized based on DSC and TGA thermal analysis of the powder to minimize porosity and warping, and they are reported in Table 3.

### 2.6. Saddle Characterization

The assessment of the modular saddle’s performance was carried out using a combined approach that integrated an experimental mechanical cyclic test of the printed structures with FEM simulations. This approach allowed for a comprehensive evaluation of the saddle’s behaviour under different loading conditions and facilitated the optimization of the design.

The main aim was to characterize the material properties of the sintered TPU, assess the influence of different lattice geometries on mechanical performance, and validate FEM predictions using real experimental data.

#### 2.6.1. Density Measurement

To refine the material’s properties further, the density of the printed TPU was measured on a 3D-printed cube (10 × 10 × 10 mm^3^) using an analytical balance model Kern ACJ 300-4M (Arroweld Italia S.P.A., Zanè (VI), Italy) equipped with a universal density set model YDB-03 (Arroweld Italia S.P.A., Zanè (VI), Italy).

#### 2.6.2. Mechanical Testing

The mechanical properties of the 3D-printed TPU samples were characterized through uniaxial tensile testing, in accordance with ASTM D638-08 [35]. Standard dog-bone specimens were prepared and tested under controlled displacement conditions to capture the stress–strain behaviour. Five specimens (n = 5) were tested under identical loading conditions, and the results are reported as mean values ± SD. This test was preliminary for the next FEM simulation to find the real elastic modulus of the TPU samples.

Moreover, a cyclic compressive test was conducted on several lattice structures by varying geometrical parameters as reported in Table 1 to evaluate the influence of different lattice geometries on mechanical performance, also following the work of Major et al. [38]. Six samples were printed, using Gyroid samples as described in the previous paragraph. The mechanical tests were performed using a Litem pneumatic modular test system MF-P-BYC (DRC S.r.l. Ancona, Italy) equipped with a 100 KN load cell, a Real Time RTC controller, and a PID-controlled system for precise force and displacement control. The system allowed for both stress-controlled and strain-controlled tests, enabling the evaluation of the samples under cyclic loading conditions. The tests were performed at a controlled temperature of 23 ± 2 °C and a relative humidity of 50 ± 5%, with samples conditioned for at least 16 h prior to testing. For each lattice configuration, three specimens (n = 3) were tested under identical loading conditions, and Δε% are reported as mean values ± SD. Given the saddle’s modular configuration in which the user’s weight is primarily supported by the two posterior sections, the tests were designed to replicate the maximum load expected on a single element. The applied force was calculated assuming a user body mass of 95 kg, with half of the resultant static load distributed over one rear module, scaled according to the contact surface of the experimental specimens. Stress-controlled fatigue tests were performed by applying a cyclic force between 10 N and 100 N at 1 Hz for 10,000 cycles, corresponding to high-cycle fatigue conditions typically experienced during rehabilitation training.

Hysteresis energy calculation (stress–strain data) was performed. For displacement-controlled tests reported as stress–strain loops, the mechanical energy dissipated per cycle was computed as the closed-loop integral of stress with respect to strain, which yields the energy per unit volume (J·m^−3^) as shown in Equation (3).(3)$$E_{cycle,vol}=\oint \sigma d\varepsilon$$

Numerically, the integral was evaluated by the trapezoidal integration of the measured stress–strain loop. The total energy dissipated by the specimen was obtained multiplying by the specimen volume V: E_cycle_ = E_ccycle,vol_∙V. Results are reported either as energy density (J·m^−3^) or as total energy (J) and are presented as mean ± SD across replicates (n = 3). To ensure that the laboratory tests realistically reproduced the loading conditions experienced by the saddle during rehabilitation cycling, the applied forces were derived from a representative body mass of 95 kg. The total body weight was calculated as W = m∙g = 95 kg × 9.81 m/s^2^ = 931.95 N. If approximately 50% of the body weight is supported by the saddle in the seated position (f_saddle_ = 0.5) and the load is distributed over two posterior modules (n_module_ = 2), the static force per module is F_module_ = W∙fs_addle_/n_modules_ = 233 N.

The module–specimen correlation was established by area scaling between the contact surface of one saddle module and that of the printed specimen. Considering a module contact area of A_module_ = 9.73 × 10^−3^ m^2^ (corresponding to the effective support area the designed saddle) and a specimen contact area of A_specimen_ = 1.26 × 10^−3^ (Ø40 mm sample), the equivalent static force on the specimen has been calculated using Equation (4).F_specimen_ = F_module_ × (a_specimen_/A_module_) = 30 N(4)

To account for dynamic effects occurring during cycling and load fluctuations, a conservative dynamic amplification factor *k*_dyn_ = 3 was applied, leading to a peak load *F*_specimen,peak_ = 90 N. Consequently, the experimental cyclic range adopted in this study (10–100 N) encompasses both the quasi-static representative value and plausible dynamic peaks experienced during cycling.

The evolution of displacement under cyclic loading was monitored to quantify the accumulation of permanent deformation. To further investigate the mechanical response of the lattice structures under alternative loading conditions, additional cyclic compression tests were performed in displacement-controlled mode. Briefly, cyclic compression tests were also carried out under displacement-controlled conditions. Tests were conducted at 5, 7, and 10 mm-imposed displacements, corresponding to nominal strains (ε) of approximately 0.25, 0.35, and 0.50, respectively. The evolution of the hysteresis energy per cycle (area of the stress–strain loop) was monitored to evaluate the energy dissipation stability. Representative hysteresis trends are provided, each applied for 1000 cycles at 1 Hz. During testing, the evolution of the compressive stress was continuously recorded to evaluate the progressive variation in load-bearing capacity with the number of cycles. This approach provided complementary information on the stiffness degradation and energy dissipation of the lattice structures when subjected to repeated deformations of fixed amplitude, thereby replicating alternative loading scenarios relevant to rehabilitation cycling. Overall, the combination of stress-controlled and displacement-controlled fatigue tests allowed for a more comprehensive characterization of the cyclic behaviour of the TPU lattices, capturing both the accumulation of permanent deformation under load-controlled conditions and the progressive stiffness degradation under imposed displacement amplitudes.

#### 2.6.3. FEM Simulation and Validation

FEM simulations were carried out to reproduce the experimental mechanical response of the lattice geometries and to validate the predictive capability of the numerical model. Each lattice configuration was modelled in Autodesk Fusion 360 software version 2604.1.25 using tetrahedral 10-node elements to accurately represent the complex periodic surfaces of the gyroid geometry, while material parameters (elastic modulus, Poisson’s ratio, and density) were derived from uniaxial tensile tests and density measurements on bulk TPU specimens. The mesh consisted of approximately 120,000 tetrahedral elements with an average size of 0.8 mm, and a mesh-sensitivity analysis confirmed displacement convergence within ±3%.

Instead of applying a uniform load across all geometries, the applied forces were tailored for each configuration. This choice was made to reproduce the experimental displacement ranges recorded during compression tests, thereby ensuring that the simulated deformation fields were directly comparable with the laboratory results. In practice, for each lattice, the applied load was adjusted so that the global deformation predicted by the model matched the corresponding experimental condition. This approach allowed for a one-to-one validation of the FEM predictions against the measured mechanical response of each geometry, rather than relying on a purely theoretical load case. By combining the experimental results with the FEM predictions, the simulation model was proven to be reliable and could be used for further design optimization. The FEM analysis allowed for an in-depth evaluation of the different lattice configurations and helped in selecting the most appropriate design for the modular saddle.

#### 2.6.4. Cost Model

A cost model was developed to estimate the production cost of the modular rehabilitation saddle using additive manufacturing technologies starting from a previously developed model adopted for a 3D-printed back brace [39]. The approach integrates and adapts cost modelling frameworks from the literature to account for both the metallic and polymeric components fabricated via Laser Powder Bed Fusion (LPBF) and Selective Laser Sintering (SLS), respectively [40,41]. The model parameters included printer type, materials, and saddle geometry.

The overall cost (C_saddle_ [€]) was estimated by the accumulation of four sub-costs: machine purchase cost (P), machine operation cost (O), material cost (M), and labour cost (L), as shown in Equation (5).C_saddle_ = P + O + M + L(5)

The build time (T_b_) includes both the active printing time (T_s_) and additional overhead (T_d_) such as layer change delays, warm-up time, and post-processing setup, as shown in Equation (3). These parameters were evaluated separately for the two geometries considered (G8_0.2 and G8_0.3), with individual components defined as follows:T_b_ = T_d_ + T_s_ = {L_p_ × (T_pre_ + T_post_) + T_startup_} + T_s_(6)

The purchase cost (P) was allocated based on the useful machine life (Y_life_) and its expected usage for saddle production, considering that such specialized equipment is typically shared among multiple applications or research purposes. According to Yim et al., machines are assumed to have a 95% machine uptime during their useful life (Y_life_) in order to calculate purchase cost per hour [41]. This cost was normalized per hour of usage and multiplied by the build time and was defined as follows:P = (T_b_ × P_c_)/(0.95 × 24 h/day × 365 days/year × Y_life_)(7)

The machine operation cost (O) was computed as the product of operation rate (C_o_ [€/h]) and build time (T_b_), as shown in Equation (8). C_o_ is an empirically determined constant, correlated to maintenance, consumption of utilities, and cost of space.O = T_b_ × C_o_(8)

Material cost (M), Equation (9), was estimated based on the TPU powder used for the ergonomic covering. A support factor (K_s_) was applied where necessary to account for additional material used for support structures or non-recyclable powder losses. The other factors N, ν, and ρ represent the number of parts, part volume [cm^3^], and material density [kg/cm^3^], respectively.M = K_r_ × K_s_ × N × ν × ρ × C_m_(9)
where C_m_ is the cost per kilogram of powder, ν is the part volume, ρ is the material density, and K_s_ is the support factor.

Finally, labour cost (L) was derived from the estimated manual time required for pre-processing (e.g., model setup, powder loading), post-processing (e.g., cleaning, surface finishing, threading), and assembly of the modular parts. Labour cost was obtained by multiplying the time required by the average technician hourly rate as reported in Equation (10).L = T_l_ × C_l_(10)

## 3. Results

### 3.1. Thermal and Mechanical Preliminary Results

Thermal and mechanical screening was first performed to compare the suitability of PP and TPU powders for SLS. Thermogravimetric analysis (TGA) was carried out to analyze the thermal stability of the TPU and PP powders. Figure 3b shows that the decomposition of TPU started from 300 °C and two stages of decomposition were observed due to the presence of dual segments in the polymer matrix. TGA revealed that PP undergoes rapid mass loss above 350 °C, whereas TPU maintained thermal stability up to 380 °C.

DSC tests were performed to determine the melting point of the two powders as well as to calculate the degree of curing of the printed part (Figure 3a). The glass window (GW) was defined as the temperature difference between the onset of recrystallization and onset of melting. TPU is composed of soft and stiff segments, and the T_g_ can be tailored by controlling the composition of each segment [42]. As illustrated in Figure 3a, TPU powders exhibited two glass transitions: T_g_ of hard segment at 101 °C, and T_g_ of the soft segments at −17 °C. The melting of TPU powders occurred around 160 °C; meanwhile, the recrystallization peak was around 82 °C and the entire recrystallization ranged from 150 °C to 100 °C. Thus, the removal chamber temperature needs to be set just below the offset of recrystallization to secure a stable arrangement of polymer segments after solidification and minimize the dimensional shrinkage of printed parts. In comparison, PP shows a relatively smaller GW with a broad melting peak around 141 °C. A sharp crystallization peak at 100 °C with a high crystallization enthalpy is clearly observed, leading to a strong tendency to shrinkage during cooling. These characteristics were reported to be the reason why PP is more challenging to be processed by SLS than TPU.

Even if PP is more challenging to process with SLS compared to TPU, to further support the material selection process for the ergonomic lattice covering, a preliminary mechanical characterization was performed on both TPU and PP. Three-dimensional-printed samples with G6_02 lattice geometry were fabricated by SLS using both TPU and PP polymers and subjected to uniaxial compressive tests to evaluate the elastic behaviour, strength, and deformation capacity of the two polymers.

Results reported in Figure 3c demonstrated distinct mechanical profiles for each material. TPU exhibited a maximum stress of 0.036 MPa at 15% strain for G6_02 geometry, confirming its elastomeric behaviour and ability to undergo large deformations without failure. The lattice elastic modulus (E*) calculated in the linear region was approximately 0.233 MPa.

In contrast, PP reached a significantly higher peak of 0.350 MPa at the same strain, showing a more rigid behaviour. Moreover, the lattice elastic modulus from the linear region was of 3.66 MPa, indicating that PP showed a more “rigid” behaviour if compared with TPU. Given the application’s requirements, namely cyclic compliance, energy absorption, and comfort under load, TPU was selected as the preferable material for the proposed application. Its superior ductility and elastic response make it better suited for fatigue-driven and comfort-critical applications like rehabilitation cycling.

### 3.2. Porosity Analysis and Fatigue Resistance

In all configurations, the measured porosity is slightly lower than the theoretical CAD value. This discrepancy is well documented in the SLS processing of thin-walled TPMS structures and is mainly attributed to manufacturing-induced geometric deviations. In particular, the combined effects of laser over-sintering, thermal diffusion beyond the nominal contour, and the laser spot size being comparable to the nominal wall thickness lead to an effective increase in strut thickness. Additional contributions arise from melt-pool dynamics and particle coalescence, which densify the structure locally.

The fatigue resistance of the polymeric lattice structures was evaluated to replicate the cyclic loading conditions representative of rehabilitation cycling. To identify the most suitable lattice configuration for the rehabilitation saddle, three quantitative performance criteria were defined prior to data analysis: (i) permanent strain (Δε) after 10,000 cycles lower than 1%, ensuring dimensional stability and recovery during prolonged use; (ii) apparent stiffness within a comfort-compatible range, comparable to that of conventional ergonomic saddles (5–15 MPa equivalent modulus); and (iii) stable energy dissipation, evaluated from the hysteresis area of cyclic curves, indicating consistent elastic recovery without mechanical degradation.

Representative stress–strain curves for all the 3D-printed samples are reported in Figure 4. The progressive increase in strain from 0.20 to 0.30 for G6_0.2 confirmed the occurrence of cyclic softening and permanent strain accumulation. Similar results are also reported for G8_0.2 and G10_0.2, even if the last one exhibited a more rigid behaviour with lower permanent strain accumulation between the first and the last cycle. The shape of the hysteresis loops provides insight into the energy dissipation characteristics. The 30% infill structures maintained wider, more consistent loops throughout the testing, indicating a stable energy absorption capacity. In contrast, the 0.2 mm wall thickness lattice structures exhibited a progressive narrowing of the loops, suggesting a deterioration of the material’s ability to dissipate energy with continued cycling.

The results for all lattice configurations are summarized in Table 2. Samples with 0.2 mm wall thickness exhibited the highest permanent engineering strain percentage (Δε%) after 10,000 cycles at 1 Hz, up to 11.22 ± 0.48% after 10,000 cycles, indicating an insufficient stiffness to sustain long-term loading. Conversely, specimens with 0.3 mm wall thickness demonstrated superior stability, with the deformation limited to 0.98 ± 0.34%. Furthermore, cell size played a significant role in modulating fatigue performance. Larger cells (10 mm) enhanced load distribution and reduced deformation (5.4%), whereas smaller cells (6 mm) promoted stress localization, resulting in accelerated damage accumulation. Conversely, smaller cells led to higher stiffness, which could enhance performance in specific applications but at the cost of reduced comfort. Based on these findings, a target deformation range of 0–1% after 10,000 cycles was defined as the most suitable for rehabilitation cycling applications, ensuring both comfort and structural integrity during prolonged use. The results revealed that the Gyroid structure with 8 mm cell size and 0.3 mm wall thickness provided an optimal balance between stiffness and comfort if compared with the other geometries by limiting permanent deformation to 0.77 ± 0.05% while preserving compliance. As the cell size increased, the stiffness of the lattice decreased, improving comfort but compromising mechanical resistance. Hysteresis energy (computed as the closed-loop area of the stress–strain cycle) was evaluated for cycles 1 and 10,000 for each considered gyroid geometries and results are reported in Table 4. Energy densities were in the order of 10^2^–10^3^ J·m^−3^; the corresponding energy per specimen ranged between 0.008 J and 0.084 J. Most geometries exhibited only small changes in hysteresis energy after 10,000 cycles (e.g., G8_0.3: −1.9%; G6_0.3: +4%), while some configurations showed larger variations (G6_0.2: +23%, G8_0.2: −11%). These trends confirm that, for the selected optimal geometry (G8_0.3), energy dissipation remains essentially stable over extended cyclic loading.

A more detailed analysis of the deformation mechanisms under strain-controlled cycling was conducted, as shown in Figure 5, Figure 6 and Figure 7. These figures depict the hysteretic behaviour and the evolution of the compressive stress over 1000 cycles for prescribed strain levels of 0.25, 0.35, and 0.50 for all the 3D-printed Gyroid structures. The cyclic curves shown in Figure 5 represent the typical behaviour of a single specimen under load-controlled testing and are included for illustrative purposes only.

The G6 lattice structure reveals a distinct behaviour between the two infill densities, as shown in Figure 5. The lower density structure (G6_0.2) exhibits a significant decay in the maximum stress sustained over the cycles, particularly at higher strain (0.35 and 0.50), indicating pronounced cyclic softening and material degradation. Conversely, the higher density structure demonstrates remarkable stability, with minimal stress decay, confirming its superior resistance to fatigue damage even under larger imposed strains. The hysteresis loops for G6_0.3 are also notably wider, indicating a greater energy dissipation per cycle that is a desirable attribute for shock absorption.

Figure 6 shows a similar trend, further validating the critical role of wall thickness also for the G8 lattice structures. The G8_0.2 configuration undergoes a progressive reduction in load-bearing capacity, especially evident at a higher strain (0.5) displacement level. In contrast, the G8_0.3 sample maintains a consistent stress response across all cycling stages, underscoring its structural integrity. In the light of these findings, the G8_0.3 geometry appears to offer a beneficial compromise, allowing for a substantial deformation (evident in the wide hysteresis loops) and ensuring that the lattice retains its elastic recovery and does not accumulate damage rapidly.

The analysis on G10 finally illustrates the effect of a larger cell size, as reported in Figure 7. While both densities show some degree of stress relaxation, it is less severe than in their smaller-cell counterparts. The larger cell size inherently provides higher compliance, leading to lower overall stress levels for the same global displacement. This translates to a reduced driving stress for damage initiation and propagation. The G10_0.3 structure exhibits the most stable response with minimal variation in peak stress, aligning with the data in Table 2, where it shows the lowest total permanent deformation (4.4%). The strain-controlled fatigue tests (Figure 5, Figure 6 and Figure 7) provide crucial insights beyond the permanent deformation metrics from stress-controlled tests (Table 2). They demonstrate that:Infill percentage (wall thickness) is the primary driver for fatigue stability. The 0.3 mm wall thickness consistently outperformed the 0.2 across all cell sizes, showing minimal stress decay and superior structural integrity under cyclic deformation.Cell size modulates stiffness and stress distribution. Larger cell sizes (10 mm) enhance compliance and reduce stress concentrations, leading to improved fatigue life. Smaller cells (6 mm) increase stiffness but are more susceptible to stress localization and damage accumulation if not coupled with sufficient density.G8_0.3 offers an optimal balance. It combines adequate compliance (for comfort) with exceptional fatigue resistance and energy dissipation, as evidenced by its stable hysteretic response and low permanent deformation. This configuration effectively mitigates cyclic softening while maintaining the elastic recovery necessary for long-term use in rehabilitation cycling.

This comprehensive fatigue analysis confirms that the G8_0.3 geometry is the most suitable design, achieving the target of limiting permanent deformation to below 1% while ensuring a reliable performance under dynamic loading conditions. These findings establish a design framework for tailoring the mechanical behaviour of additively manufactured saddles, where controlled deformation is essential to guarantee user comfort without compromising structural integrity.

### 3.3. FEM Simulation and Mechanical Test Validation

FEM simulations were conducted to complement the experimental fatigue assessment and to gain insight into the stress–strain distribution within the gyroid lattice structures. Prior to performing the numerical simulations, experimental tests were conducted to determine the material parameters required for the FEM model. Uniaxial tensile tests on SLS-TPU dog-bone specimens provided the stress–strain response and enabled the extraction of the relevant mechanical properties. The elastic modulus was found to be 8.144 ± 1.016 MPa, the ultimate tensile strength 2.313 ± 0.187 MPa, and the elongation at break 1.013 ± 0.099, confirming the highly ductile and compliant nature of TPU. These results were consistent with the expected nonlinear elastic behaviour of thermoplastic elastomers and were used to define the constitutive law of the material within the FEM framework. In addition, the apparent density of the printed TPU samples was experimentally measured and used as an input parameter to account for the relative density of the lattice structures. This difference can be attributed to the sintering process which resulted in higher packing density and reduced porosity in the material. The accurate representation of both stiffness and density was considered essential for a reliable prediction of stress distribution and deformation mechanisms under cyclic loading. The FEM simulations were conducted on representative gyroid lattice geometries with varying cell size and infill percentage, corresponding to the experimental configurations. Boundary conditions replicated the compression tests, with the upper platen applying displacement or load control depending on the experimental setup, as shown in Figure 8.

To validate the numerical approach, FEM predictions were systematically compared with the results of quasi-static compression tests on selected lattice specimens. Representative FEM analysis results are reported in Figure 8, while the quantitative deviation between numerical and experimental results is summarized in Table 5. The FEM model employed a linear elastic approximation of TPU with E = 8.144 ± 1.016 MPa, σ_max_ = 2.313 ± 0.187 MPa, ε_max_ = 1.013 ± 0.099, ν = 0.3 ± 0.01, and density of 0.89 ± 0.03 g/cm^3^. While TPU exhibits viscoelastic behaviour, this simplified model was adopted to enable a comparative stiffness evaluation between lattice configurations. The model is therefore not predictive of time-dependent or hysteretic effects. Overall, the FEM model reproduces the global trend observed experimentally: denser lattices require larger simulated reaction forces to reach comparable deformations and show smaller relative deviations between the simulation and experiment, while highly porous samples display larger mismatches. The best agreement is observed for G8_0.3, with a discrepancy of approximately 5% between simulated and experimental displacement, whereas the largest deviations occur for the most porous configurations where they appear to be up to ~20–30% in some cases. To quantify the agreement between the experimental and numerical results, the absolute percentage deviation was calculated according to the following equation:$$∆\%=(\frac{x_{exp}-x_{FEM}}{x_{FEM}})\cdot 100$$

Experimental displacement values are reported as the mean ± SD of three independent specimens for each configuration. The Δ(%) column expresses the absolute percentage deviation between the average experimental and FEM-predicted displacement.

These quantitative differences reflect a combination of modelling assumptions and intrinsic differences between the as-designed geometry and the printed parts. In particular, the numerical model used a homogenized, linear-elastic description of the TPU and idealized geometry imported from the CAD file. In contrast, the physical specimens exhibit manufacturing-induced geometric imperfections (local variations in strut thickness, surface roughness, partially fused powder, and local densification) and possible through-thickness anisotropy in the mechanical response. For highly porous gyroids these local effects (for example, local buckling of thin walls, contact between neighbouring struts, and localized yielding) strongly influence the deformation mode but are not fully captured by the present simplified constitutive and geometric model.

Despite these limitations, the FEM predictions capture the global stiffness ranking of the architectures and reproduce the orders of magnitude of the load–displacement response. For medium-density lattices (0.3 mm wall thickness) the model is sufficiently predictive to support design decisions, whereas for very open structures it should be considered indicative and used in combination with experimental verification.

### 3.4. Modular Saddle 3D Printing and Cost Model

Following the design and characterization phases, the modular saddle was physically manufactured to demonstrate the feasibility of the proposed concept. The metallic support, produced by LPBF, was designed as a standardized base capable of hosting interchangeable lattice modules. The polymeric covering, fabricated by SLS using the optimized processing parameters reported in Table 3, represents the customizable element of the system, allowing for the geometry and stiffness of the saddle surface to be tailored to the specific needs of each patient.

As illustrated in Figure 9, the modular design enables the addition or replacement of lattice segments, which can be fabricated on demand and easily integrated into the metallic frame. This approach provides a straightforward pathway to adapt the saddle to user-specific anatomical and functional requirements. To further verify its practical applicability, the complete modular saddle, comprising both the metallic base and the TPU lattice covering, was mounted on a cycle ergometer. This preliminary implementation confirmed that the modular design facilitates personalization while ensuring compatibility with rehabilitation equipment.

Building on this proof of concept, a cost model was developed to evaluate the economic sustainability of manufacturing a customizable polymeric covering through SLS. As previously discussed, the metallic support is intended as a reusable standard component, while the TPU lattice is the variable part requiring patient-specific customization and thus representing the main cost driver. To further evaluate the feasibility of manufacturing rehabilitation saddles via additive processes, a cost model of the polymeric 3D-printed part was developed. The detailed cost estimation for the selected G8_0.3 configuration is reported in Table 6.

Among the six geometries investigated, only the most promising configuration, the G8_0.3, was selected for detailed cost estimation, as it demonstrated the best compromise between mechanical stability, comfort, and fatigue resistance. The cost estimation was also evaluated for the most porous one, G8_0.2, to understand how the geometric features can affect the final cost of the saddle. This approach allowed the model to focus on the configuration most likely to reach real clinical adoption. It should be noted that the cost analysis was restricted to the polymeric covering, as the metallic support is intended to remain a standardized and reusable component across different patients. In contrast, the TPU lattice covering represents the customizable element of the modular design and therefore constitutes the critical variable in terms of production cost and scalability. The unit cost was computed as the sum of machine purchase/depreciation, operation/overhead, material, and labour, following widely used AM cost formulations (overall cost = P + O + M + L) and operation rate (C_o_) treatment as in Yim & Rosen (C_o_ × build time) [41]. For mixed builds, batch-level costs were allocated to parts using a build-share factor consistent with parallel production in powder bed AM as reported by Ruffo et al., which emphasizes equitable allocation under varying packing densities and layer counts [40]. An operation rate (C_o_) equal to €4.75·h^−1^ is derived from utility tariffs and preventive maintenance allocation specific to our SLS cell (excludes depreciation). This corresponds to the “operation rate” concept used as factory overhead in AM models by Yim et al. [41]. K_s_ is used to account for the cost of using more materials while constructing support structures. K_s_ is 1 in the absence of a support structure. Similarly, K_r_, which represents the recycling factor has been set equal to 1 because it was supposed that all the non-sintered material is reused after printing. In the cost model, the density (ρ = 0.65 g/cm^3^) refers to the bulk density of the TPU powder feedstock, which is used to calculate the material cost based on the total powder volume involved in the SLS process. As reported in Table 6, the total production cost of the TPU covering for the G8_0.3 configuration amounts to 53.00 €, while the slightly less dense variant G8_0.2 shows a comparable value of 52.30 €. The machine purchase cost was normalized over the expected lifetime of the equipment, resulting in a negligible contribution (<1 € per part). The largest share of the cost was associated with machine operation, calculated as 14.13 € for a build time of ≈3 h. Material costs remained relatively low (5.51–6.21 €), thanks to the limited volume of TPU powder required for the covering. Labour accounted for a fixed contribution of 33.00 €, corresponding to pre-processing, setup, and post-processing operations.

Overall, the analysis highlights that labour and machine operation dominate the cost structure, while material expenses contribute only marginally. This is consistent with previous findings regarding the small-scale AM production of medical devices. Importantly, the estimated cost per saddle covering falls within a competitive range, especially considering the high degree of customization and modularity enabled by the AM process.

These results suggest that additive manufacturing of modular rehabilitation saddles can be economically viable in small-to-medium batch production, where the added value of patient-specific adaptation outweighs the higher operating costs compared to mass-produced conventional saddles. Further details on the sensitivity analysis and robustness of the cost model are reported in the Appendix A.

Reliable data on the actual costs of saddles handcrafted in rehabilitation settings are not available in the scientific literature because most of the literature focuses on the design and optimization of the saddle [43] or on the seating and positioning [44,45]. Nevertheless, as a market reference, several commercial services for customized cycling saddles report prices in the range of 250–650 €, depending on the level of personalization and the technology adopted [21,22,23]. These values are not directly comparable to a clinical context, since commercial prices also include design services, individual fitting, logistics, and retail margins. However, they provide a useful order of magnitude to position the estimated cost of our polymeric component. Moreover, some reviews on assistive seating emphasize the clinical benefits of customization without providing detailed economic data, confirming the scarcity of published evidence on actual rehabilitation saddle costs [46].

## 4. Discussions

The findings of this study demonstrate the successful integration of additive manufacturing techniques in developing an ergonomic, modular bicycle saddle tailored for rehabilitation purposes. The mechanical evaluation revealed that the G8_0.3 lattice structure exhibited optimal performance, achieving only a 0.3% permanent deformation after 10,000 fatigue cycles. The superior load distribution and energy absorption characteristics make it particularly suitable for rehabilitation applications, where repetitive loading and long-term durability are critical. The results obtained from the fatigue test and FEM simulations highlight the complementary role of experimental and numerical approaches in the optimization of lattice-based structures for rehabilitation applications. Fatigue experiments demonstrated that both the infill percentage and cell size strongly influence long-term durability, with low-density lattices undergoing excessive permanent deformation and high-density structures ensuring improved stability.

The FEM/experimental comparison provides two important practical insights for the use of FEM in lattice design. First, model fidelity matters more for highly porous architectures: when struts are thin and the structure is open, local phenomena (buckling, contact, manufacturing variability) dominate the response and require a nonlinear geometric/material model and, ideally, as-built geometry to be modelled. Second, the chosen constitutive description of TPU strongly affects accuracy: a linear elastic model calibrated on a small-strain tensile test captures the initial stiffness but cannot reproduce the pronounced nonlinearity, viscoelasticity and cyclic softening observed experimentally. Consequently, the smallest deviations (≈5–10%) for denser lattices in Table 5 reflect cases in which the global response is dominated by the bulk stiffness and global bending of the unit cells, which the simplified model can approximate; larger deviations for porous samples indicate the onset of local instability and nonlinearity that the model currently omits.

Methodologically, tailoring the applied FEM loads to reproduce the experimental displacement for each geometry is a pragmatic choice for one-to-one model validation because it ensures that the simulated deformation field corresponds to the laboratory condition used for the measurement. Finally, the combined FEM/experimental framework remains valuable: FEM efficiently screens design trends and identifies promising configurations (G8_0.3), while targeted experiments validate critical cases. Future work should focus on improving the material model (hyperelastic/viscoelastic plus cyclic damage), incorporating geometric imperfections (as-built scans) and enabling nonlinear buckling/contact analyses for the most porous architectures, which will reduce the remaining simulation–experiment gaps and enable fully predictive digital design workflows.

From an economic perspective, focusing the cost model on the polymeric covering is consistent with the modular design concept of the saddle, where the metallic frame acts as a fixed structural base and only the TPU lattice requires patient-specific customization. This highlights that the main cost drivers in real-world applications will be associated with the production of the variable, patient-adapted component rather than the reusable metallic support. Moreover, our cost model, focused on the replaceable TPU covering, appears competitive when compared with available market benchmarks for customized saddles, while a dedicated clinical-economic analysis (including technician time, materials, and reimbursement policies) will be required for a definitive comparison with traditional handcrafted solutions in rehabilitation centres.

It is worth noting that a usability and acceptability study on healthy volunteers has already received ethical approval from the CNR Ethics Committee (protocol n. 0316288, 26/08/2025) and will be the subject of our forthcoming work. This clinical-oriented investigation will complement the present mechanical and modelling analyses, providing essential insights into the practical adoption of the modular saddle in rehabilitation contexts.

## 5. Conclusions

This study presented the design and development of a modular rehabilitation saddle manufactured by combining LPBF for the metallic support and SLS for the polymeric covering. Preliminary material screening identified TPU as the most suitable polymer, owing to its superior mechanical response under cyclic loading and its favourable processability, as confirmed by thermal analyses.

Fatigue experiments on lattice structures demonstrated that both the cell size and infill percentage are critical parameters for long-term durability. The optimal configuration (8 mm cell size, 0.3 mm wall thickness) exhibited limited permanent deformation (≈0.3% after 10,000 cycles), achieving an effective balance between compliance and stability. FEM simulations further validated these findings, showing good agreement with experimental data for medium-to-high density structures, and confirming their potential as predictive tools for design optimization.

The cost model highlighted the feasibility of producing customized components through additive manufacturing, where the modular design and digital workflow enable efficient adaptation to patient-specific needs while maintaining competitive production costs in small-batch scenarios.

Overall, this research demonstrates that the integration of advanced AM processes, targeted material selection, and combined FEM/experimental evaluation can yield high-performance customizable solutions for rehabilitative medical devices. The methodology outlined here can be extended to other biomedical applications requiring lightweight, fatigue-resistant, and ergonomically optimized components. In conclusion, the modular design was effectively validated through the fabrication of a complete saddle, which was mounted on a cycle ergometer to demonstrate its functional applicability. This proof-of-concept highlights how additive manufacturing can deliver patient-specific solutions by combining standardized metallic components with customizable polymeric lattices, thus merging clinical adaptability with economic feasibility. Finally, a usability and acceptability study on healthy volunteers, already approved by the CNR Ethics Committee (protocol n. 0316288, 26/08/2025), will represent the next step of this research, paving the way for clinical validation of the modular saddle concept. The study will gather feedback from participants, who will use the customized saddle for 10 min and compare it to a standard saddle, also used for 10 min. Opinions regarding comfort and ergonomics will be collected through ad hoc questions.

## Figures and Tables

**Figure 1 materials-18-05242-f001:**
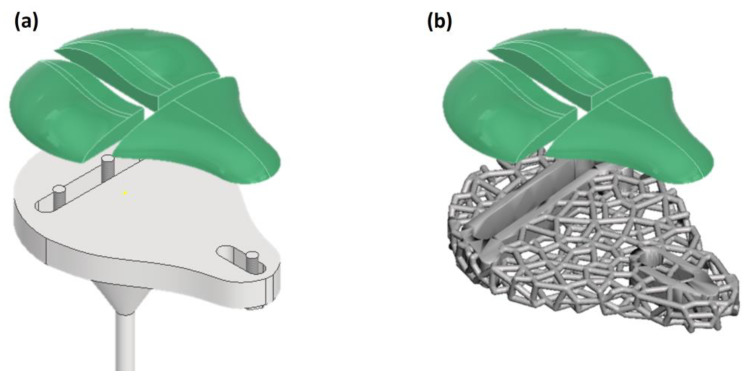
Modular saddle design. (**a**) CAD model of the saddle consisting of two main components: a metallic support (grey part) and a polymeric ergonomic covering (green part) that can be adjusted based on the patient’s specific needs; (**b**) Voronoi-lattice-optimized metal structure.

**Figure 2 materials-18-05242-f002:**
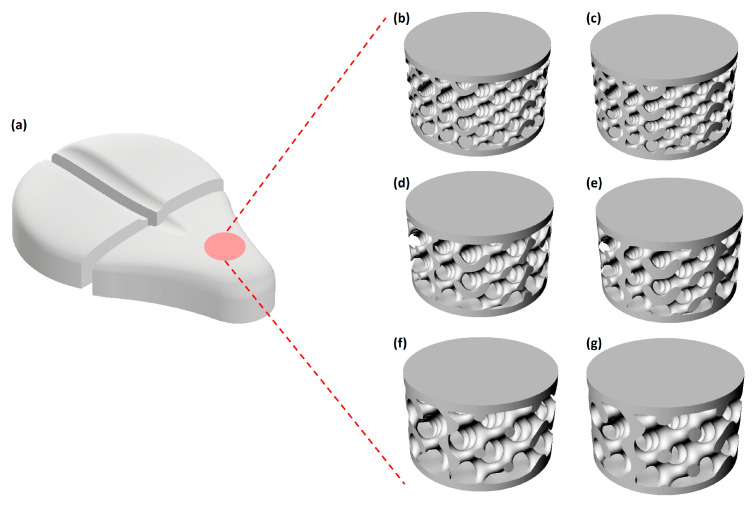
(**a**) Modular polymeric saddle design by using different gyroid lattice structures (**b**) G6_0.2, (**c**) G6_0.3, (**d**) G8_0.2, (**e**) G8_0.3, (**f**) G10_0.2, and (**g**) G10_0.3.

**Figure 3 materials-18-05242-f003:**
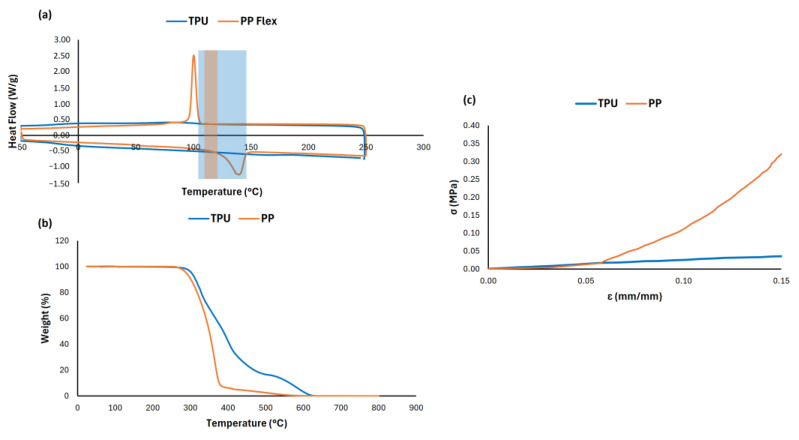
Preliminary characterization of the 3D-printed samples reporting DSC (**a**), TGA (**b**), and stress–strain curves of uniaxial compressive tests on G6_0.2 (**c**). Shaded area represent the Glass window for TPU (blue) and PP Flex (orange).

**Figure 4 materials-18-05242-f004:**
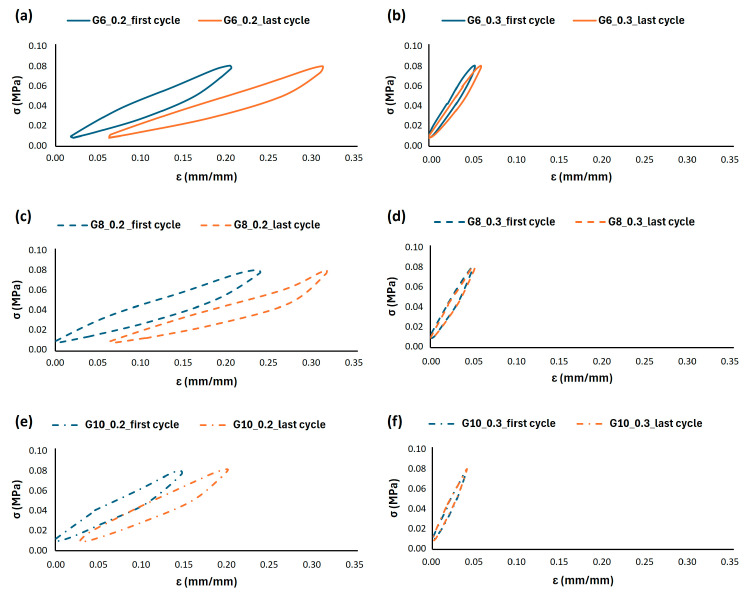
Representative cyclic stress–strain curve in load control for (**a**) G_6_0.2, (**b**) G6_0.3, (**c**) G8_0.2, (**d**) G8_0.3, (**e**) G10_0.2 and (**f**) G10_0.3. The curve illustrates the typical behaviour of a single specimen; averaged results are reported in Table 2 and Table 4.

**Figure 5 materials-18-05242-f005:**
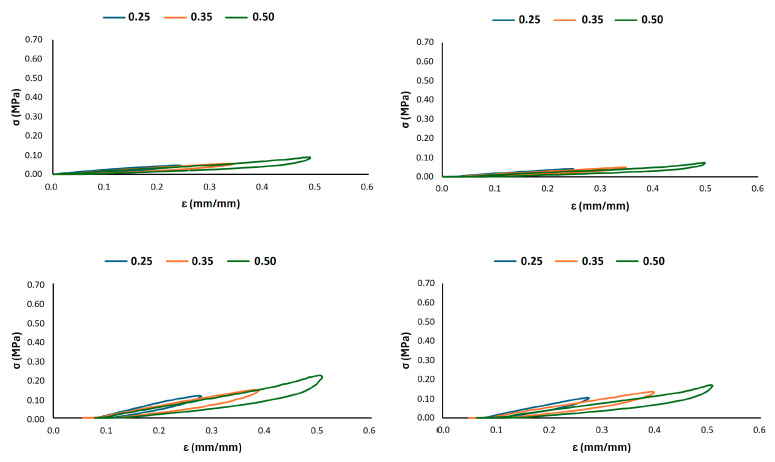
Strain-controlled fatigue tests at different displacement levels for G6_0.2 (top) and G6_0.3 (bottom) configurations, after the 1st (left) and 1000th (right) cycle hysteresis loops.

**Figure 6 materials-18-05242-f006:**
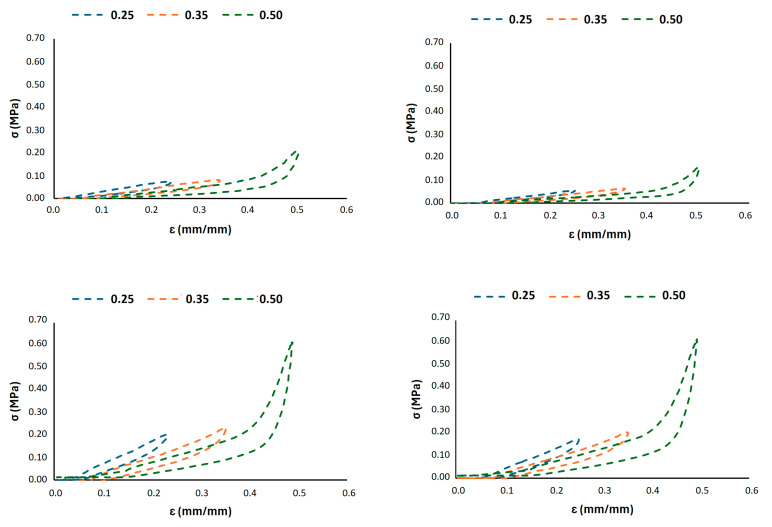
Strain-controlled fatigue tests at different displacement levels for G8_0.2 (top) and G8_0.3 (bottom) configurations, showing the first (left) and last (right) cycle hysteresis loops.

**Figure 7 materials-18-05242-f007:**
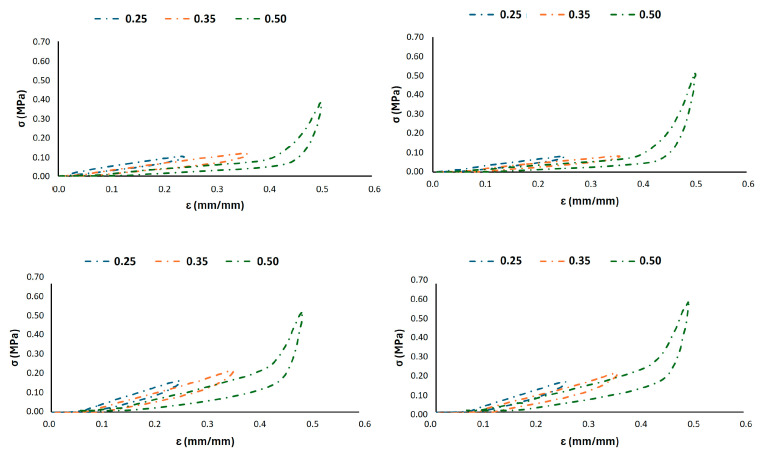
Strain-controlled fatigue tests at different displacement levels for G10_0.2 (top) and G10_0.3 (bottom) configurations, showing the first (left) and last (right) cycle hysteresis loops.

**Figure 8 materials-18-05242-f008:**
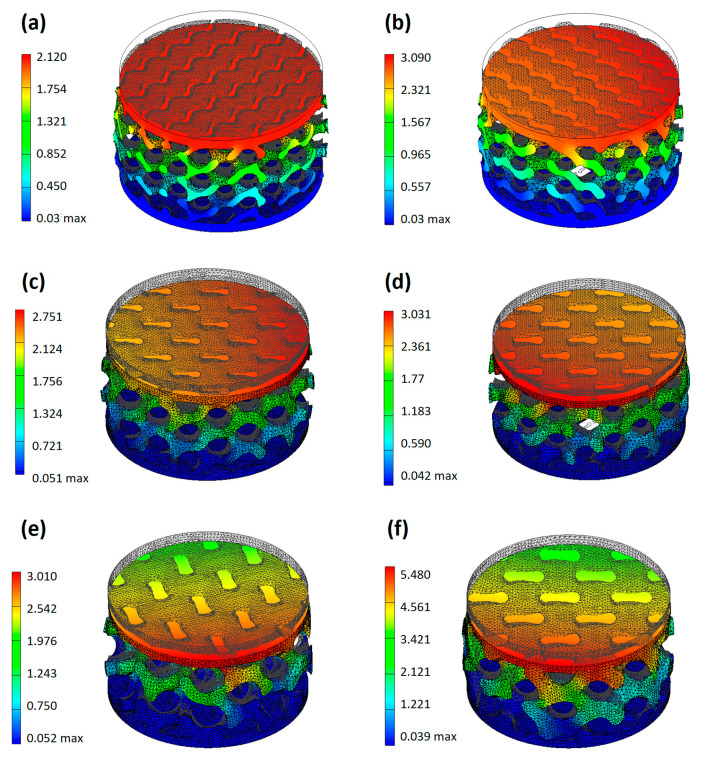
FEM analysis of (**a**) G6_0.2, (**b**) G6_0.3, (**c**) G8_0.2, (**d**) G8_0.3, (**e**) G10_0.2, and (**f**) G10_0.3.

**Figure 9 materials-18-05242-f009:**
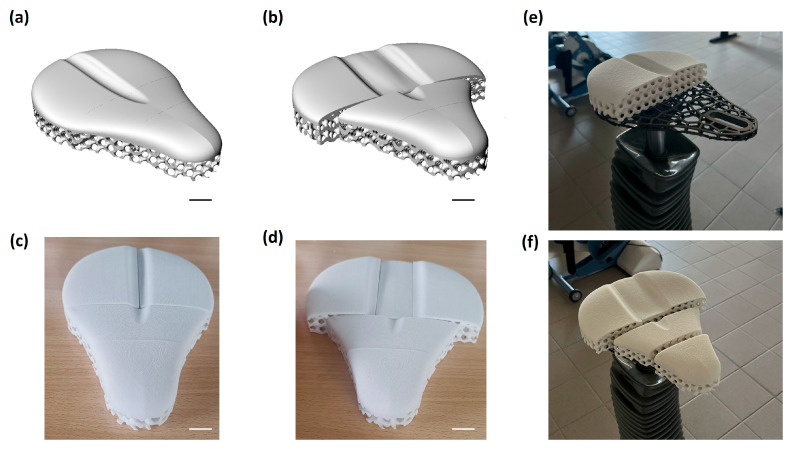
A modular saddle manufactured according to the optimized LPBF and SLS parameters. The metallic base (**e**) acts as standardized support, while the TPU lattice covering (**c**,**d**,**f**) can be replaced or adapted through additively manufactured modules, enabling patient-specific customization by acting on the CAD file of the saddle (**a**,**b**) (Scale bars represent 2 mm.).

**Table 1 materials-18-05242-t001:** Optimized LPBF process parameters for AISI 316L.

Parameters	Values
Scanning strategy	Meander
Laser spot size [µm]	30
Platform temperature (°C)	25
Atmosphere	Ar (O_2_ < 100 ppm)
Hatch distance [µm]	90
Point distance [µm]	50
Thickness layer [µm]	50
Power [W]	200
Scanning time [µs]	70

**Table 2 materials-18-05242-t002:** Comparison of fatigue resistance: cell size, wall thickness, theoretical and measured porosity, and permanent deformation after 10,000 cycles for SLS-TPU gyroid lattices. Values are expressed as the mean of three specimens per configuration (n = 3).

Sample	Cell Size (mm)	Wall Thickness (mm)	Theoretical Porosity (%)	Measured Porosity (%)	Δε (%)
G6_0.2	6	0.2	79.62	75.01 ± 2.16	11.22 ± 0.48
G6_0.3	6	0.3	69.46	66.92 ± 1.29	0.95 ± 0.06
G8_0.2	8	0.2	79.69	75.06 ± 0.19	8.94 ± 0.88
G8_0.3	8	0.3	69.79	64.50 ± 0.37	0.77 ± 0.05
G10_0.2	10	0.2	79.59	76.51 ± 0.27	5.12 ± 0.23
G10_0.3	10	0.3	69.52	63.00 ± 0.17	0.98 ± 0.34

**Table 3 materials-18-05242-t003:** Optimized SLS process parameters for TPU and PP powders.

Parameters	TPU	PP
Laser Power (W)	6.3	2.8
Scan Speed (mm/s)	2400	2400
Scan Spacing (µm)	100	100
Layer Height (µm)	100	100
Atmosphere	Air	Air
Powder Bed Temp (°C)	120	135

**Table 4 materials-18-05242-t004:** The computed energy densities (E, J·m^−3^) for cycles 1 and 10,000 for each 3D-printed geometry.

Sample	E_1_ (J·m^−3^)	E_10000_ (J·m^−3^)	ΔE (%)	E_1_ (J)	E_10000_ (J)
G6_0.2	2.6 ± 0.1 × 10^3^	3.1 ± 0.2 × 10^3^	+23%	0.064	0.079
G6_0.3	5.8 ± 0.2 × 10^2^	6.1 ± 0.3 × 10^2^	+4%	0.015	0.015
G8_0.2	3.3 ± 0.1 × 10^3^	3.0 ± 0.2 × 10^3^	−11%	0.084	0.074
G8_0.3	5.2 ± 0.2 × 10^2^	5.1 ± 0.1 × 10^2^	−1.9%	0.013	0.013
G10_0.2	2.0 ± 0.1 × 10^3^	2.3 ± 0.4 × 10^3^	+15%	0.050	0.058
G10_0.3	3.5 ± 0.4 × 10^2^	3.3 ± 0.3 × 10^2^	−7.6%	0.0088	0.0082

**Table 5 materials-18-05242-t005:** Comparison between experimental and FEM displacement results (mean ± SD) and corresponding absolute percentage deviation (Δ%).

Sample	Force (N)	FEM Displacement (mm)	Experimental Displacement (mm)	Δ (%)
G6_0.2	55	2.12	3.01 ± 0.34	29.50
G6_0.3	176	3.09	3.81 ± 0.27	22.30
G8_0.2	60	2.75	3.52 ± 0.24	21.87
G8_0.3	182	3.03	3.19 ± 0.16	5.01
G10_0.2	75	3.01	3.25 ± 0.22	7.38
G10_0.3	235	5.48	5.90 ± 0.19	7.77

**Table 6 materials-18-05242-t006:** A detailed cost breakdown of the proposed cost model for producing a TPU saddle with the G8_0.2 and G8_0.3 lattice structures.

**Unit cell length [mm]**	**8**	
**Wall thickness [mm]**	**0.2**	**0.3**
Build time—T_b_ [h]	2.97	2.97
Purchase price—P_c_ [€]	12,000	12,000
Expected life—Y_life_ [years]	7	7
**Machine cost—P [€]**	**0.66**	**0.66**
Operation rate—C_o_ [€ /h]	4.75	4.75
**Operative cost—O [€]**	**14.13**	**14.13**
Recycling factor (K_r_)	1	1
Support material factor—K_s_	1	1
Number of parts—N	1	1
Part volume—ν [cm^3^]	164.48	185.47
Material rate per unit weight—C_m_ [€/kg]	51.5	51.5
Material density—ρ [g/cm^3^]	0.65	0.65
**Material cost—M [€]**	**5.51**	**6.21**
Labour time—T_l_ [h]	1	1
Labour rate—C_l_ [€/h]	33	33
**Labour cost—L [€]**	**33.00**	**33.00**
**Overall Cost—C [€/part]**	**52.30**	**53.00**

## Data Availability

The original contributions presented in this study are included in the article/Appendix A. Further inquiries can be directed to the corresponding author.

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
