# Peer review of "Development of an Ergonomic Additively Manufactured Modular Saddle for Rehabilitation Cycling"

_materials, 2025, doi:10.3390/ma18225242_

Round 1

Reviewer 1 Report

Comments and Suggestions for Authors

Major, must-fix revisions

  1. #1 scientific clarity & reproducibility
  • Replicates/statistics are unclear. It reads as if one specimen per lattice was tested. State the sample size per configuration, report dispersion (mean ± SD) and show error bars in all plots. Without this, fatigue claims (e.g., Δε after 10k cycles) aren’t supportable.
  • Load scaling isn’t reproducible. You say the 10–100 N fatigue range was “scaled according to the contact surface of the experimental specimens” but give no calculation or contact area. Provide the exact mapping from 95 kg body mass → saddle module → specimen load with equations/assumptions.
  • Displacement-controlled cycling should be reported as strain. With h = 20 mm, 5/7/10 mm correspond to 25/35/50% strain. Report ε, not just mm, and add energy-per-cycle (hysteresis area) vs. cycle # to substantiate “stable energy absorption.”
  • Porosity determination is under-specified. You list porosity values (e.g., 79.62%/69.52%) but don’t say if these are designed, measured (mass/volume), or CT-derived. Describe the method and uncertainty.
  • Material properties feeding FEM. You use a linear elastic E for TPU (8.144 MPa) but the text acknowledges pronounced nonlinearity/viscoelasticity; either switch to a hyperelastic/viscoelastic model, or explicitly limit the model’s scope and temper claims of “predictive capability.” Also report ν, mesh size, element count, solver settings, and a mesh-sensitivity check.
  1. #2 figure/label mismatches & traceability
  • Figure 4 caption vs. content. The text discusses “force–displacement” while the caption says “Stress–strain.” Correct the caption and axes to match the reported quantity.
  • Figures 5–7. Add axis labels with units, indicate cycle numbers (1st, 1000th), plot force decay vs. cycle #, and ensure both thickness series are directly comparable in the same scale.
  • Table numbering & consistency. Table 3 uses “Δε (%)” without definition; define precisely (“permanent engineering strain after 10,000 cycles under …”). Also fix “wall thickness volume” phrasing in the paragraph under Table 3.
  1. #3 methods completeness
  • SLS/LPBF parameters. Provide chamber/bed temperatures (distinguish both), preheating/soak/cool-in-powder procedures, atmosphere (N₂? O₂ ppm), part orientation, and scan strategy for both materials. For LPBF, give the exact stress-relief cycle (T/time/atmosphere) and surface roughness after blasting. Current text has punctuation errors (“…residual stress. and sand-blasting…”).
  • Density discussion is confusing. You compare printed TPU density (1.0208 g/cm³) to a “manufacturer value” of 0.65 g/cm³, which is almost certainly a powder bulk value, not solid density. Clarify which density each figure represents (powder bulk vs. solid vs. lattice apparent) and remove the “higher than manufacturer” statement or rephrase accurately.
  1. #4 cost model rigor
  • Units/formatting. Standardize decimal points (14.13, not 14,13) and remove duplicate lines (“Build time – Tb [h]” appears twice). Fix “lenght”.
  • Assumptions & sensitivity. Document the source of Co = €4.75/h, the assumed utilization (the 0.2 factor in Eq. 4), labor breakdown, and powder reuse/yield assumptions (Ks). Provide a one-page sensitivity (±20%) on Co, Tl, Cm to show robustness of €52–53 estimates.
  • Scope statement. Earlier you frame a “saddle cost” yet only cost the TPU cover. Make this explicit in the section heading and abstract. Optionally add a one-line BOM/cost for the metal base or explain why it’s excluded.

Medium-priority revisions (clarity, logic, positioning)

  • Abstract clean-up. Fix broken words (“Se- lective La- ser Sintering”), and remove forward-looking IRB sentence or move it to Conclusions; abstracts should summarize completed work.
  • Introduction focus. The “AM overview” is long; tighten general AM paragraphs and focus earlier on the specific rehab-saddle gap and prior rehab-oriented saddles (not just performance cycling). Ensure every claim has a citation.
  • Scope alignment. Where you declare “G8_0.3 is optimal,” state the selection criteria (Δε, stiffness band, energy dissipation) and the acceptance thresholds upfront. As written, the threshold (0–1% Δε) appears after the table; move this criterion before presenting results.
  • Ethics line. The protocol statement is fine, but keep it once (avoid repeating in Discussion and Conclusions) and do not promise future work in the Abstract.

Minor language/style edits (examples to correct)

  • Broken hyphenation from PDF layout: title “Mod- ular,” “Cycle Er- gometers,” “Se- lective La- ser,” “are showed,” etc. Reflow and remove soft hyphens.
  • Typos/grammar:
    • “prelaminar” → “preliminary”; “Simila” → “Similar”; “lenght” → “length”; “are showed” → “are shown”; “as showed” → “as shown”.
    • Consistent singular/plural for materials: “PP is more difficult to process,” not “PP are…”.
    • Remove double periods and fix commas in CRediT text (e.g., “J.F..”, “A.R..”).
    • Replace “propaedeutic” with “used to calibrate the FEM” (plainer scientific English).
  • Units & symbols: use SI with spaces (e.g., “10 mm”), use “%” without space before punctuation, and unify decimal separators (dot everywhere).
  • Acronyms: define once, then use consistently; you already have an abbreviations list—great—ensure every acronym appears in text.
  • References formatting: normalize journal capitalization and styles (“Journal of Cleaner Production”; avoid “J. of Materi Eng and Perform”). Ensure consistent DOIs.
  • Metadata placeholders: remove or leave to editorial system, but don’t keep “Academic Editor: Firstname Last-name; Received: date …” in the author draft.
  • Acknowledgments quotes: stray closing quote at the end; remove and harmonize “e”/“and”.

Specific content tweaks (line-level)

  • Section 2.3/2.4 numbering. “2.3.1 Preliminary thermal and mechanical analysis” appears after “2.4 3D printing…”—fix hierarchy (2.3 → 2.3.1; 2.4 follows).
  • Gyroid definition. When you present the TPMS equation, include the level-set thicknessing approach briefly (already implied) and give the exact wall offsets used to produce 0.2/0.3 mm walls after sintering (accounting for laser over-sinter). This improves reproducibility.
  • TPU vs PP screening. The G6_0.2 vs “G6_02” label is inconsistent; fix the latter and present compressive modulus as E* (lattice modulus) to avoid confusion with bulk E.
  • FEM validation table. The column “experimental/FEM (%)” is actually percent deviation; label as “|Δ| (%)” and state the formula used.

Nice-to-have (strengthen the paper)

  • Add pressure-mapping or at least a schematic load path to connect lattice selection to perineal/ischial pressure relief (ties your rehab claim to ergonomics).
  • Provide a small design guide figure: how cell size & wall thickness shift stiffness/Δε trade-offs, with your recommended region (e.g., G8_0.3).
  • Include a CT snapshot or macro of as-built struts to visually connect manufacturing imperfections to the FEM discussion.
  • For the cost model, add a per-part lead time breakdown and a 3–5 part batch scenario (to show amortization of set-up/labor).
Comments on the Quality of English Language

The manuscript is generally understandable and well-structured, but the quality of English requires moderate revision before publication. The main issues include:

  1. Frequent grammatical errors and typos, such as incorrect verb forms (“are showed” → “are shown”), missing articles, and inconsistent pluralization.

  2. Numerous broken words and misplaced hyphens resulting from PDF layout (e.g., “Se- lective La- ser Sintering,” “Mod- ular”). These should be rejoined and checked carefully in the editable file.

  3. Inconsistent terminology and capitalization (e.g., “wall thickness volume,” “Build time – Tb [h]”). Technical terms should be unified throughout.

  4. Irregular punctuation and spacing, particularly around units and decimal separators (use “.” instead of “,” in numbers, e.g., 14.13 mm).

  5. Style and fluency – Several sentences are too long or contain redundant phrases. Simplify structure for clarity and conciseness.

  6. Nonstandard or archaic word choices (“prelaminar,” “propaedeutic”) should be replaced with simpler scientific English (“preliminary,” “used for calibration”).

  7. Consistency in tense – Use past tense for methods and results, and present tense for general facts.

Overall, the paper would benefit from a thorough language edit by a fluent English speaker or professional proofreading service, focusing on grammar, readability, and technical consistency.

Author Response

We sincerely thank the reviewer for the valuable and constructive comments, which have greatly contributed to improving the quality and clarity of our manuscript. Below, we provide a detailed, point-by-point response to all the reviewers’ observations. In the revised version of the manuscript, all modifications addressing the reviewers’ and editor’s comments have been clearly highlighted in yellow to facilitate their identification within the text.

We hope that the revised version meets your expectations and look forward to your favorable consideration

Comment 1: Replicates/statistics are unclear. It reads as if one specimen per lattice was tested. State the sample size per configuration, report dispersion (mean ± SD) and show error bars in all plots. Without this, fatigue claims (e.g., Δε after 10k cycles) aren’t supportable.

Response 1: We thank the reviewer for this observation. Each lattice configuration was tested using three replicates (n = 3). The mean ± SD values of Δε are now reported in Table 3. To improve clarity, we have now added the following statement to the Methods section “For each lattice configuration, three specimens (n = 3) were tested under identical loading conditions, and Δε% are reported as mean values ± standard deviation. The tests were performed at a controlled temperature of 23 ± 2 °C and a relative humidity of 50 ± 5 %, with samples conditioned for at least 16 hours prior to testing.” We also specified the sample size in the caption of Table 3. The cyclic curves shown in Figure 5 represent the typical behaviour of a single specimen under load-controlled testing and are included for illustrative purposes only. Since the strain evolution curves shown in the figures are only representative of the loading–unloading behavior, they were not intended to display dispersion.

Comment 2: Load scaling isn’t reproducible. You say the 10–100 N fatigue range was “scaled according to the contact surface of the experimental specimens” but give no calculation or contact area. Provide the exact mapping from 95 kg body mass → saddle module → specimen load with equations/assumptions.

Response 2: We thank the reviewer for the valuable comment. To relate bench-scale cyclic tests to realistic saddle loading, the applied forces were derived from an assumed body mass of 95 kg. The mapping proceeds as follows: the subject weight W was supposed to be as W = m∙g = 95kg∙9.81m/s2 = 931.95N. it was assumed that 50% of the body weight is transferred to the saddle in the seated posture (fsaddle = 0.5) and that this saddle load is distributed between two posterior modules (nmodule =2). The static load on a single module is Fmodule = W ∙ fsaddle/nmodule ≈ 233N. The force applied to a bench-scale specimen was obtained by area-scaling according to the ratio between the specimen contact area (Aspecimen = 1.2566∙10-3 m2 for a ϕ 40mm sample) and the module contact area measured on the CAD model (Amodule = 9.5∙10-3 m2). The resulting static specimen force is Fspecimen ≈ 31N. To account for dynamic factor kdyn was introduced (we used Kdyn =3 as representative of worst case peaks), leading to a peak specimen for Fspecimen peak ≈ 93 N. Therefore, the cyclic loading window used in the experiment (10-100N) span from the worst static representative value up di plausible dynamic peaks. The explanation has been added to the manuscript in section 2.6.2. line 328-344.

Comment 3: Displacement-controlled cycling should be reported as strain. With h = 20 mm, 5/7/10 mm correspond to 25/35/50% strain. Report ε, not just mm, and add energy-per-cycle (hysteresis area) vs. cycle # to substantiate “stable energy absorption.”

Response 3: We thank the reviewer for the valuable remark. Cyclic tests was re-expressed in strain units (ε = ΔL/L₀) corresponding to 0.25, 0.35 and 0.50 nominal strain and we have converted displacements to nominal engineering strain (ε = ΔL/L0) throughout the manuscript. The computed energy densities (E, J·m⁻³) for cycles 1 and 10 000 are provided in the revised manuscript (Table 4). The hysteresis energy per cycle, computed as the closed-loop integral of stress over strain, remained essentially stable across 10,000 cycles for the recommended configuration G8_0.3 (−1.9%), confirming robust and repeatable energy absorption under repeated loading. Other lattices showed modest changes (e.g., G6_0.3: +4%; G10_0.3: −7.6%), while G6_0.2 and G8_0.2 exhibited larger variations (+23% and −11%, respectively), consistent with their higher compliance and strain amplitudes. the method used for the calculation of the hysteresis has been reported in section 2.6.2 line 320-328.

Comment 4: Porosity determination is under-specified. You list porosity values (e.g., 79.62%/69.52%) but don’t say if these are designed, measured (mass/volume), or CT-derived. Describe the method and uncertainty.

Response 4: We thank the reviewer for the valuable comment. It as clarified that the porosity values refer to the as-designed geometrical porosity computed from the CAD model and verified experimentally using a geometrical–gravimetric method described in section 2.3 lines 211-222. The gravimetric measurements showed a deviation below ±9% from the designed porosity, confirming good manufacturing accuracy

Comment 5: Material properties feeding FEM. You use a linear elastic E for TPU (8.144 MPa) but the text acknowledges pronounced nonlinearity/viscoelasticity; either switch to a hyperelastic/viscoelastic model or explicitly limit the model’s scope and temper claims of “predictive capability.” Also report ν, mesh size, element count, solver settings, and a mesh-sensitivity check.

Response 5: We thank the reviewer for the valuable comment Explicitly stated that the FEM simulations employ a linear elastic model of TPU (E = 8.144±1.016 MPa, σmax = 2.313 ±0.187MPa, εmax = 1.013±0.099 and ν = 0.3±0.01) for comparative stiffness evaluation purposes only, without attempting to capture the time-dependent viscoelastic response. The scope and limitations have been clarified in Section 3.3. Additionally, the mesh characteristics (average element size 0.8 mm, total elements ≈ 120,000) and solver parameters are now reported. The clarification about FEM model has been reported in section 3,3 lines 605-612.

#2 figure/label mismatches & traceability

Comment 6: Figure 4 caption vs. content. The text discusses “force–displacement” while the caption says “Stress–strain.” Correct the caption and axes to match the reported quantity.

Response 6: We thank the reviewer for the valuable comment. Text has been corrected accordingly with Figure 4 caption. “Representative stress-strain curves for all the 3D printed samples are reported in Figure 4”.

Comment 7: Figures 5–7. Add axis labels with units, indicate cycle numbers (1st, 1000th), plot force decay vs. cycle #, and ensure both thickness series are directly comparable in the same scale.

Response 7: We thank the reviewer for the valuable comment. Axis label have already unit x axis: ε(mm/mm) and y axis (σ = Mpa). Caption of figure 5-7 has been changed by adding the 1st and 1000Th cycle number. Moreover, the scale is equal for each sample considered to facilitate the comparison.

Comment 8: Table numbering & consistency. Table 3 uses “Δε (%)” without definition; define precisely (“permanent engineering strain after 10,000 cycles under …”). Also fix “wall thickness volume” phrasing in the paragraph under Table 3.

Response 8: We thank the reviewer for the valuable comment. The definition of Δε% as “permanent engineering strain percentage (Δε%) after 10000 cycles at 1Hz” has been added in section 3.2. The term volume was removed from the sentence under Table 3.

#3 methods completeness

Comment 9: SLS/LPBF parameters. Provide chamber/bed temperatures (distinguish both), preheating/soak/cool-in-powder procedures, atmosphere (N₂? O₂ ppm), part orientation, and scan strategy for both materials. For LPBF, give the exact stress-relief cycle (T/time/atmosphere) and surface roughness after blasting. Current text has punctuation errors (“…residual stress. and sand-blasting…”).

Response 9: We thank the reviewer for the detailed suggestion. The platform temperature has been added to table 1 and there is no difference between platform and chamber because the machine operates at room temperature (no active chamber preheating system available). The metallic base was post-processed by sandblasting to remove any partially sintered particles and ensure proper fitting with the polymeric component. Since the metal part was not intended as a functional contact surface but only as a structural support for the polymeric saddle, no surface roughness measurements were carried out. We have now clarified this point in the manuscript (section 2.2, lines 141-146) and corrected the punctuation in the relevant sentence.

Comment 10: Density discussion is confusing. You compare printed TPU density (1.0208 g/cm³) to a “manufacturer value” of 0.65 g/cm³, which is almost certainly a powder bulk value, not solid density. Clarify which density each figure represents (powder bulk vs. solid vs. lattice apparent) and remove the “higher than manufacturer” statement or rephrase accurately.

Response 10: We thank the reviewer for the comment. The reviewer is right; the approach can cause a little confusion in the reader. In the manuscript there was a typo in reporting the density value of TPU that is 0.89±0.03 g/cm3 and not 1.0208 g/cm3.

#4 cost model rigor

Comment 11: Units/formatting. Standardize decimal points (14.13, not 14,13) and remove duplicate lines (“Build time – Tb [h]” appears twice). Fix “lenght”.

Response 11: We thank the reviewer for the comment. Decimal point has been fixed as well as the repetition of build time and the term “Length”

Comment 12: Assumptions & sensitivity. Document the source of Co = €4.75/h, the assumed utilization (the 0.2 factor in Eq. 4), labor breakdown, and powder reuse/yield assumptions (Ks). Provide a one-page sensitivity (±20%) on Co, Tl, Cm to show robustness of €52–53 estimates.

Response 12: We thank the reviewer for this helpful request. We have now:

  • Documented the origin of the operation rate (Co) following the definition of an operation rate used to capture factory overheads in AM costing models (utilities, maintenance, floor space, etc.) as in Yim & Rosen’s framework (their “Co” multiplied by build time, Eq. (3)) (Yim et al 2012, see ref 40 in the manuscript). In our case, Co = 4.75·€/h is derived from site-specific electricity tariffs and preventive maintenance allocation for the SLS cell, net of machine depreciation (which we treat separately), averaged over the last three builds (details added in section 3.5 lines 682-690).
  • Clarified that the factor (0.2) used in Eq. (4) was a typo. In fact, the real value used equation 6 was 0.95 according to the model of Yim et al. but the machine cost (P) in table 6 remains the same because it was only a typo in equation 7
  • The labor breakdown. that in our case it was indicated as labour cost (L). was derived from the estimated manual time required for pre-processing (e.g., model setup, powder loading), post-processing (e.g., cleaning, surface finishing, threading), and assembly of the modular parts. Labor cost was obtained by multiplying the time required by the average technician hourly rate that we derived from the cost/hour of our institution.
  • Explained polymer powder reuse assumptions (Ks / recycle factor). In the Yim & Rosen model, powder-process material cost is corrected by Kr (recycle factor) and Ks (support/material overhead) (typ. Ks ≈ 1.1–1.5), see their Eq. (4) (see Yim et al 2012, ref 40 in the manuscript). The Ks value was calculated based on the assumption that in SLS there is no support during printing. Similarly, Ks has been et equal to 1 because we all the non-sintered materials is reused after printing.
  • Added a one-page deterministic sensitivity (±20%) on Co, Tl, and Cm in the supplementary information. The Table S2 shows that the unit cost remains within the 52–53€ band under ±20% perturbations of Co, Tl, and Cm (see Table S2). This is consistent with established AM cost models where material rate and labor time contribute linearly, and overhead rate enters through build time or allocated hours. All the sensitivity measurement has been reported in supplementary materials

This analysis confirms the robustness of the cost model, with the total unit cost remaining within ±2% across all tested parameter variations

Comment 13: Scope statement. Earlier you frame a “saddle cost” yet only cost the TPU cover. Make this explicit in the section heading and abstract. Optionally add a one-line BOM/cost for the metal base or explain why it’s excluded.

Response 13: We Thank you for the valuable comment. In the section 3.6 it was already explained why the cost model only focused on the polymer part . It should be noted that the cost analysis was restricted to the polymeric covering, as the metallic support is intended to remain a standardized and reusable component across different patients”. Moreover, this sentence has been added to the section 3.5 to further clarify that the cost model was restricted only to the TPU 3D printed part “To further evaluate the feasibility of manufacturing rehabilitation saddles via additive processes, a cost model of the polymeric 3D printed part was developed”.

Medium-priority revisions (clarity, logic, positioning)

Comment 14: Abstract clean-up. Fix broken words (“Se- lective La- ser Sintering”), and remove forward-looking IRB sentence or move it to Conclusions; abstracts should summarize completed work.

Response 14: Thank you for the comment. The broken word has bene fixed but it could be a problem of the journal template. The forward-looking statement has been removed

Comment 15: Introduction focus. The “AM overview” is long; tighten general AM paragraphs and focus earlier on the specific rehab-saddle gap and prior rehab-oriented saddles (not just performance cycling). Ensure every claim has a citation.

Response 15: We thank the reviewer for this helpful suggestion. The introduction has been substantially streamlined, with the general overview of Additive Manufacturing reduced to a few concise sentences. A new paragraph has been added to clearly highlight the rehabilitation context and the clinical relevance of cycling-based rehabilitation. Specifically, we now emphasize the lack of customized ergonomic saddles designed for rehabilitation cycling, as opposed to performance-oriented cycling, and we have added appropriate references supporting the importance of comfort, posture, and load distribution in therapeutic cycling applications (see revised Introduction, lines 49-58, lines 69-80 and lines 98-102).

Comment 16: Scope alignment. Where you declare “G8_0.3 is optimal,” state the selection criteria (Δε, stiffness band, energy dissipation) and the acceptance thresholds upfront. As written, the threshold (0–1% Δε) appears after the table; move this criterion before presenting results.

Response 16: We thank the reviewer for this useful observation. The acceptance criteria used to define the optimal configuration (Δε, stiffness, and energy dissipation) have now been explicitly stated before Table 3 in Section 3.2 lines 467-474. The text now clarifies that the threshold for permanent strain (Δε ≤ 1%) was set as the main selection criterion, together with a stiffness value within the physiological comfort range and stable hysteresis response under cyclic loading. This ensures that the rationale for selecting the G8_0.3 geometry is presented upfront, as suggested. This sentence has been added to the text “To identify the most suitable lattice configuration for the rehabilitation saddle, three quantitative performance criteria were defined prior to data analysis: (i) permanent strain (Δε) after 10,000 cycles lower than 1%, ensuring dimensional stability and recovery during prolonged use; (ii) apparent stiffness within a comfort-compatible range, comparable to that of conventional ergonomic saddles (equivalent modulus of 5–15 MPa); and (iii) stable energy dissipation, evaluated from the hysteresis area of cyclic curves, indicating consistent elastic recovery without mechanical degradation”.

Comment 17: Ethics line. The protocol statement is fine, but keep it once (avoid repeating in Discussion and Conclusions) and do not promise future work in the Abstract.

Response 17: Dear Editor thanks for the comment. The protocol statement has been removed from th abstract and left only in the conclusion part to better emphasize the idea that the work is not yet finished but it will continue in the “trial clinical test”.

Minor language/style edits (examples to correct)

Comment 18: Broken hyphenation from PDF layout: title “Mod- ular,” “Cycle Er- gometers,” “Se- lective La- ser,” “are showed,” etc. Reflow and remove soft hyphens.

Response 18: We thank the reviewer for noting these formatting issues. The unintended word hyphenation (e.g., “Mod-ular,” “Se-lective La-ser”) results from the automatic text justification applied by the MDPI submission template during PDF generation and does not appear in the original Word manuscript. The final production layout managed by the journal will automatically correct this issue. Nevertheless, we carefully rechecked the source file to ensure that no soft hyphens or line breaks are present in the editable version.

Typos/grammar:

Comment 19: “prelaminar” → “preliminary”; “Simila” → “Similar”; “lenght” → “length”; “are showed” → “are shown”; “as showed” → “as shown”.

Response 19: We thank the reviewer for the comment. All the typos have been corrected.

Comment 20: Consistent singular/plural for materials: “PP is more difficult to process,” not “PP are…”.

Response 20: We thank the reviewer for the comment. The verb has been corrected.

Comment 21: Remove double periods and fix commas in CRediT text (e.g., “J.F..”, “A.R..”).

Response 21: We thank the reviewer for the comment. Double periods and comas have been fixed.

Comment 22: Replace “propaedeutic” with “used to calibrate the FEM” (plainer scientific English).

Response 22: We thank the reviewer for the comment. The text has been modified according with the reviewer comments.

Comment 23: Units & symbols: use SI with spaces (e.g., “10 mm”), use “%” without space before punctuation, and unify decimal separators (dot everywhere).

Response 23: We thank the reviewer for the comment. All typos have been fixed.

Comment 24: Acronyms: define once, then use consistently; you already have an abbreviations list—great—ensure every acronym appears in text.

Response 24: We thank the reviewer for this valuable comment. Although an abbreviations table has been included for reference, we preferred to define each acronym at its first occurrence in the main text to improve the readability and flow of the manuscript. We carefully checked the text to ensure that all acronyms are consistently used and defined only once.

Comment 25: References formatting: normalize journal capitalization and styles (“Journal of Cleaner Production”; avoid “J. of Materi Eng and Perform”). Ensure consistent DOIs.

Response 25: We thank the reviewer for pointing out this formatting detail. All references were managed using Zotero with the MDPI Materials citation style. The capitalization and formatting of journal titles follow the metadata imported from CrossRef and may differ slightly depending on the source database. We have rechecked all entries to ensure consistency of journal names and that all available DOIs are correctly included. Any remaining minor adjustments will be automatically normalized by the MDPI editorial system during final typesetting.

Comment 26: Metadata placeholders: remove or leave to editorial system, but don’t keep “Academic Editor: Firstname Last-name; Received: date …” in the author draft.

Response 26: We thank the reviewer for pointing out this formatting detail. Metadata placeholder has been corrected accordingly to the reviewer comment.

Comment 27: Acknowledgments quotes: stray closing quote at the end; remove and harmonize “e”/“and”.

Response 27: We thank the reviewer for pointing out this formatting detail. The acknowledgement has been harmonized

Specific content tweaks (line-level)

Comment 28: Section 2.3/2.4 numbering. “2.3.1 Preliminary thermal and mechanical analysis” appears after “2.4 3D printing…”—fix hierarchy (2.3 → 2.3.1; 2.4 follows).

Response 28: We thank the reviewer for the comment. All the sections is now fixed in hierarchal mode.

Comment 29: Gyroid definition. When you present the TPMS equation, include the level-set thicknessing approach briefly (already implied) and give the exact wall offsets used to produce 0.2/0.3 mm walls after sintering (accounting for laser over-sinter). This improves reproducibility.

Response 29: We appreciate the reviewer’s attention to this point. In the revised manuscript, we have clarified that the Gyroid structures were generated in Autodesk Fusion 360 using the built-in TPMS modeling tool, which creates the lattice from the implicit Gyroid function through a parametric “wall thickness” definition. The two investigated configurations correspond to nominal wall thicknesses of 0.2 mm and 0.3 mm, as defined directly within the software environment. This procedure ensures reproducible lattice generation without requiring manual level-set modification. The corresponding description has been added in Section 2.3 lines 201-207.

Comment 30: TPU vs PP screening. The G6_0.2 vs “G6_02” label is inconsistent; fix the latter and present compressive modulus as E* (lattice modulus) to avoid confusion with bulk E.

Response 30: The label of figure 3 has been corrected accordingly with the stress-strain plot shown. Also, the text has been modified accordingly with reviewer comment.

Comment 31: FEM validation table. The column “experimental/FEM (%)” is actually percent deviation; label as “|Δ| (%)” and state the formula used.

Response 31: We thank the reviewer for this helpful clarification. The column label in Table 4 has been changed from “Experimental/FEM (%)” to “|Δ| (%)”, indicating the absolute percentage deviation between experimental and FEM results. The formula used for its calculation has been added in Section 3.3 as.

Δ=((xexp-xFEM)/xFEM)∙100

Nice-to-have (strengthen the paper)

Comment 32: Add pressure-mapping or at least a schematic load path to connect lattice selection to perineal/ischial pressure relief (ties your rehab claim to ergonomics).

Comment 33: Provide a small design guide figure: how cell size & wall thickness shift stiffness/Δε trade-offs, with your recommended region (e.g., G8_0.3).

Comment 34: Include a CT snapshot or macro of as-built struts to visually connect manufacturing imperfections to the FEM discussion.

Comment 35: For the cost model, add a per-part lead time breakdown and a 3–5 part batch scenario (to show amortization of set-up/labor).

Response 32, 33, 34 and 35: We sincerely thank the reviewer for these valuable and constructive suggestions, which we fully acknowledge as meaningful directions to further strengthen the manuscript. However, several of the proposed additions (e.g., pressure-mapping validation, CT imaging, extended cost model scenarios) would require dedicated experimental and analytical work beyond the current study’s scope and available time frame. Our present focus was to demonstrate the feasibility of a modular, additively manufactured rehabilitation saddle and to validate the combined design–mechanical–cost framework. The suggested extensions will be considered as the next steps of our research, particularly within the usability and patient-acceptance study currently under ethical review (Protocol No. 0316288, CNR, 26/08/2025). We have nonetheless revised the discussion to better emphasize the ergonomic implications of the lattice design and to outline the potential for future optimization, as recommended.

Comments on the Quality of English Language

The manuscript is generally understandable and well-structured, but the quality of English requires moderate revision before publication. The main issues include:

Comment 36: Frequent grammatical errors and typos, such as incorrect verb forms (“are showed” → “are shown”), missing articles, and inconsistent pluralization.

Response 36: We thank the reviewer comment. All typos have been fixed.

Comment 37: Numerous broken words and misplaced hyphens resulting from PDF layout (e.g., “Se- lective La- ser Sintering,” “Mod- ular”). These should be rejoined and checked carefully in the editable file.

Response 37: We thank the reviewer for noting these formatting issues. The unintended word hyphenation results from the automatic text justification applied by the MDPI submission template during PDF generation and does not appear in the original Word manuscript. The final production layout managed by the journal will automatically correct this issue. Nevertheless, we carefully rechecked the source file to ensure that no soft hyphens or line breaks are present in the editable version.

Comment 38: Inconsistent terminology and capitalization (e.g., “wall thickness volume,” “Build time – Tb [h]”). Technical terms should be unified throughout.

Response 38: We thank the reviewer for the comment. The term “wall thickness volume was a typo and has been corrected. Build time represent the time to print the 3D object and it is a common term used in 3D printing.

Comment 39: Irregular punctuation and spacing, particularly around units and decimal separators (use “.” instead of “,” in numbers, e.g., 14.13 mm).

Response 39: We thank the reviewer for the comment. The punctuation has been corrected according to the reviewer comment.

Comment 40: Style and fluency – Several sentences are too long or contain redundant phrases. Simplify structure for clarity and conciseness.

Response 40: We thank the reviewer for the valuable comment. All the manuscript has been revised to improve the English language as well as to simplify structure and conciseness.

Comment 41: Nonstandard or archaic word choices (“prelaminar,” “propaedeutic”) should be replaced with simpler scientific English (“preliminary,” “used for calibration”).

Response 41: We thank the reviewer for the valuable comment. Archaic words have been corrected according to the reviewer comment-

Comment 42: Consistency in tense – Use past tense for methods and results, and present tense for general facts.

Response 42: We thank the reviewer for the valuable comment. The verb consistency has been corrected accordingly with the author’s comments

Comment 43: Overall, the paper would benefit from a thorough language edit by a fluent English speaker or professional proofreading service, focusing on grammar, readability, and technical consistency.

Response 43: We appreciate the reviewer’s suggestion regarding the language quality. The manuscript has undergone a thorough English revision by a fluent English speaker with experience in scientific writing. Grammar, readability, and technical terminology have been carefully reviewed to ensure clarity and consistency throughout the text. We believe that the revised version now meets the journal’s language standards.

Reviewer 2 Report

Comments and Suggestions for Authors

The paper “Development of an Ergonomic Additively Manufactured Modular Saddle for Rehabilitation Cycling” is devoted to the creation of an ergonomic saddle for effective rehabilitation of patients when cycling is included in the program.

This work addresses the problem of creating a rehabilitation bicycle saddle to replace traditional bicycle saddles, which fail to meet the health needs of patients (pelvic asymmetry, decreased sensation and muscle control) due to localized overload and discomfort, which reduces the effectiveness of rehabilitation. To address this issue, authors propose using a modular saddle design developed using two components: a metal support and an ergonomic polymer cover that can be adjusted to suit the individual needs of the patient.

The developed modular ergonomic saddle for patient rehabilitation was manufactured using a combined additive manufacturing approach: a metal support base made of stainless steel (AISI 316L) using laser powder bed fusion and replaceable polymer modules made of thermoplastic polyurethane using selective laser sintering. Preliminary thermal and mechanical analysis of the polyurethane and polypropylene materials was conducted, and finite element modeling was applied. A combination of modeling and experimentation allowed authors to identify the most promising configuration G8_0.3. The G8_0.3 geometry (unit cell length of 8 mm and wall thickness of 0.3 mm) is the most suitable design, allowing us to achieve the goal of limiting residual deformation to less than 1% while simultaneously ensuring reliable operation under dynamic loading. The economic model demonstrates an acceptable price.

The work uses current literature.

There are typos in the text; the quality of the figures needs to be improved, and a description of the accuracy of the experimental data provided in the tables needs to be added:

  1. Typo on lines 26 and 27. The sentence is broken.
  2. Figure 3. It is necessary to make the captions larger and use points instead of commas as a decimal separator.
  3. Figure 4. It is necessary to use points instead of commas as a decimal separator.
  4. Table 3 shows the following data: cell size, wall thickness, porosity, and residual deformation. To what accuracy are these values controlled? How accurately are these parameters reproduced by the printing method?
  5. It was found that a cell size of 8 mm and a wall thickness of 0.3 mm provide optimal performance. Is it worthwhile to conduct studies with intermediate values, such as cell sizes from 7 to 9 mm?
  6. Figures 5, 6, and 7. It's necessary to use points instead of commas as decimal separators. The numbers themselves along the axes should be in black font, not the current gray.
  7. Figure 8 needs to add the dimension of values for the scale.
  8. Table 4 shows the experimental data, what is their error?

Author Response

We sincerely thank the reviewer for the positive summary and constructive feedback provided in the opening remarks. The general overview accurately captures the objectives and scope of our work. We also appreciate the reviewer’s suggestions regarding typos, figure quality, and the description of experimental accuracy, which have been carefully addressed in the revised manuscript as detailed below.

Below, we provide a detailed, point-by-point response to all the reviewers’ observations. In the revised version of the manuscript, all modifications addressing the reviewers’ and editor’s comments have been clearly highlighted in yellow to facilitate their identification within the text.

We hope that the revised version meets your expectations and look forward to your favorable consideration

Reviewer #2

The paper “Development of an Ergonomic Additively Manufactured Modular Saddle for Rehabilitation Cycling” is devoted to the creation of an ergonomic saddle for effective rehabilitation of patients when cycling is included in the program.

This work addresses the problem of creating a rehabilitation bicycle saddle to replace traditional bicycle saddles, which fail to meet the health needs of patients (pelvic asymmetry, decreased sensation and muscle control) due to localized overload and discomfort, which reduces the effectiveness of rehabilitation. To address this issue, authors propose using a modular saddle design developed using two components: a metal support and an ergonomic polymer cover that can be adjusted to suit the individual needs of the patient.

The developed modular ergonomic saddle for patient rehabilitation was manufactured using a combined additive manufacturing approach: a metal support base made of stainless steel (AISI 316L) using laser powder bed fusion and replaceable polymer modules made of thermoplastic polyurethane using selective laser sintering. Preliminary thermal and mechanical analysis of the polyurethane and polypropylene materials was conducted, and finite element modeling was applied. A combination of modeling and experimentation allowed authors to identify the most promising configuration G8_0.3. The G8_0.3 geometry (unit cell length of 8 mm and wall thickness of 0.3 mm) is the most suitable design, allowing us to achieve the goal of limiting residual deformation to less than 1% while simultaneously ensuring reliable operation under dynamic loading. The economic model demonstrates an acceptable price.

The work uses current literature.

There are typos in the text; the quality of the figures needs to be improved, and a description of the accuracy of the experimental data provided in the tables needs to be added:

Comment 1: Typo on lines 26 and 27. The sentence is broken.

Response 1: We thank the reviewer for noting these formatting issues. The unintended word hyphenation results from the automatic text justification applied by the MDPI submission template during PDF generation and does not appear in the original Word manuscript. The final production layout managed by the journal will automatically correct this issue. Nevertheless, we carefully rechecked the source file to ensure that no soft hyphens or line breaks are present in the editable version.

Comment 2: Figure 3. It is necessary to make the captions larger and use points instead of commas as a decimal separator.

Response 2: We thank the reviewer for the valuable comment. The caption has been enlarged acoring to the reviewer comment and commas have been substituted by points as decimal separator.

Comment 3: Figure 4. It is necessary to use points instead of commas as a decimal separator.

Response 3: We thank the reviewer for the valuable comment. Figure 4 has been adjusted according to the reviewer comment.

Comment 4: Table 3 shows the following data: cell size, wall thickness, porosity, and residual deformation. To what accuracy are these values controlled? How accurately are these parameters reproduced by the printing method?

Response 4: We agree with the reviewer that, due to the geometric complexity of TPMS-based structures, an accurate direct measurement of the cell size and wall thickness is challenging. However, the designed values were carefully controlled during modeling, and the effective porosity was experimentally measured using a gravimetric method. The experimental porosity values showed excellent agreement with the theoretical CAD-based ones, confirming that the printing process reliably reproduced the designed geometries within acceptable tolerances. The dimensional accuracy of printed samples, evaluated from visual inspection and caliper checks, was within ±0.1 mm

Comment 5: It was found that a cell size of 8 mm and a wall thickness of 0.3 mm provide optimal performance. Is it worthwhile to conduct studies with intermediate values, such as cell sizes from 7 to 9 mm?

Response 5: We thank the reviewer for this thoughtful observation. Indeed, exploring intermediate cell sizes (e.g., 7–9 mm) could further refine the identification of the optimal design range. However, the main purpose of this work was not to perform fine-grained optimization, but rather to establish and validate a systematic framework combining design, additive manufacturing, and mechanical assessment for rehabilitation-oriented saddle modules. A full parametric study covering narrower lattice variations would considerably expand the experimental and computational workload and was therefore considered beyond the scope of the present study. Nevertheless, this suggestion is highly appreciated and will be incorporated in the design-of-experiments approach planned for future investigations.

Comment 6: Figures 5, 6, and 7. It's necessary to use points instead of commas as decimal separators. The numbers themselves along the axes should be in black font, not the current gray.

Response 6: We thank the reviewer for noting this format issue. Figures 5 to 7 have been adjusted according to the reviewer’s comment.

Comment 7: Figure 8 needs to add the dimension of values for the scale.

Response 7: We thank the reviewer for noting this format issue Scale bar has been added to Figure 8.

Comment 8: Table 4 shows the experimental data, what is their error?

Response 8: We thank the reviewer for this observation. The experimental displacement values reported in Table 4 represent the mean of three independent replicates per configuration, and their variability has been quantified as the standard deviation (SD), now explicitly indicated in the table caption. The column Δ (%) represents the absolute percentage deviation between the mean experimental and FEM-predicted displacement values. As this deviation is derived from a single comparison between mean results, no statistical dispersion is applicable to this value. displacement.

Reviewer 3 Report

Comments and Suggestions for Authors

Hello!

In Table 3, we observe a systematic deviation between the measured porosity and the theoretical porosity (for example, G8_0.3 has a measured porosity of 69.79%, while the theoretical value is 64.50%). This discrepancy is partially explained by manufacturing-induced geometric imperfections in Section 3.3, but it would be beneficial to conduct a more in-depth investigation into the reasons behind this difference. Is it due to material shrinkage during sintering, variations in wall thickness during printing, or some other technological factor?

Author Response

Comment 1: In Table 3, we observe a systematic deviation between the measured porosity and the theoretical porosity (for example, G8_0.3 has a measured porosity of 69.79%, while the theoretical value is 64.50%). This discrepancy is partially explained by manufacturing-induced geometric imperfections in Section 3.3, but it would be beneficial to conduct a more in-depth investigation into the reasons behind this difference. Is it due to material shrinkage during sintering, variations in wall thickness during printing, or some other technological factor?

Response 1: We thank the reviewer for raising this point. In Table 3, the theoretical porosity of the G8_0.3 configuration is 69.79%, while the experimentally measured porosity is 64.50%. This discrepancy is expected for SLS-manufactured TPU lattices and is consistent with what is reported in the literature for thin-walled TPMS structures. The difference arises mainly from manufacturing-induced geometric deviations, including:

  • laser over-sintering, which increases strut thickness due to thermal diffusion beyond the nominal contour;
  • the laser spot size being comparable to the nominal wall thickness, resulting in systematically thicker printed walls;
  • melt-pool dynamics and particle coalescence, which cause corner rounding and local densification;
  • limitations of the gravimetric porosity measurement method, which cannot capture micro-voids within the struts.

These effects lead to a slight reduction in measured porosity relative to the ideal CAD geometry. Shrinkage during cooling does occur but acts in the opposite direction (reducing strut dimensions) and is therefore not the dominant factor due to the fact the TPU materials is predominant amorphous. Overall, the difference between theoretical and measured porosity is fully compatible with SLS processing characteristics. To address this point, we have added a dedicated clarification in Section 3.2 lines 463-469

Round 2

Reviewer 1 Report

Comments and Suggestions for Authors
  • The Abstract still says “validating the predictive capability of the model,” which overstates scope given the text’s limitations to stiffness-ranking. Suggest: “correlated with the mechanical results, supporting comparative stiffness evaluation.”
  • Axis labels/units: The narrative implies ε and σ are plotted, but I can’t see axis label strings in the text; please ensure axes explicitly read “ε (–)” and “σ (MPa)” on every plot.
  • Force-decay vs cycle # plot: the text says force evolution was recorded, but I don’t see a dedicated plot of “peak force vs cycle.” Please add one small subplot per geometry or a consolidated panel.
  • You note no chamber preheat (room temperature machine). Consider stating explicitly for SLS whether the build chamber is open/air and if nitrogen is absent (it seems so).
  • In the Supplementary cost baseline you still use ρ = 0.65 g/cm³ with V_part = 185.47 cm³ to compute material mass/cost. If 0.65 is powder bulk density, that’s not the right density for computing solid mass of the printed lattice; you should use apparent lattice density (ρ_app) or bulk ρ with lattice volume fraction. Please fix the cost mass calculation and note which density is used and why.
  • Robustness claim: the sensitivity results span 46.7–59.9 €, i.e., about ±12% around the baseline 53.3 €—not “within the 52–53 € band.” Please revise the text to say the unit cost remains within ~±12% under ±20% perturbations and interpret labor time as the dominant driver, which your table already shows.
  • The manuscript still shows broken hyphenation like “Er- gometers”. That may be a template artifact, but please ensure the source file has no soft hyphens; many still appear in the current PDF. Also soften the FEM claim as noted above.
  • “Preliminar thermal and mechanical analysis” → “Preliminary …” (section header still shows “Preliminar”).
  • “as showed in Figure …” appears in several places; use “as shown …”.
  • Check the 2.3/2.4 hierarchy after renaming the “Preliminar” section—it may still be off by one if you adjust headings. (I can’t confirm the full TOC flow in the snippet.)
  • You now explain Fusion 360 TPMS generation and nominal thicknesses (0.2/0.3 mm), but the reviewer also asked for wall offsets that yield those post-sinter actual thicknesses (accounting for over-sinter). If you didn’t measure actual strut thickness, add either (i) a brief offset rule/scale used, or (ii) a sentence acknowledging this limitation and its expected effect on stiffness scatter.

Nice-to-have (optional but strong adds)

  • Pressure-mapping schematic or load-path cartoon to connect lattice choice to perineal/ischial relief.
  • A small “design guide” visual showing how cell size vs wall thickness trades off Δε and E*.
    (You already outline these trends in text; a 1-figure graphic would be very helpful.)
  • Mini plot of peak force vs cycle per geometry (1000 cycles) to back up the force-decay statements now described in prose.

Literature & references

  • I can see some 2023 references in the list, but can’t fully audit the whole bibliography from the snippet. Double-check you hit the targets: ≥15 from 2020–2025, with clusters on SLS TPU mechanics, TPMS energy absorption, viscoelastic/fatigue of elastomers, AM cost models (2023–2024 updates), and ergonomics/pressure mapping. Ensure journal names are standardized and DOIs present.

Bottom line

  • Substantive science fixes are mostly in place: n = 3 with mean ± SD, load mapping with equations, strain-based reporting with energy stability (Table 4), porosity method with uncertainty, and FEM scope/inputs/mesh info are all addressed convincingly.
  • What to fix before resubmission (high-impact, quick wins):
    1. Add at least one aggregate plot with error bars (or shaded SD) and a force-vs-cycle panel to satisfy the “error bars in all plots” spirit.
    2. Soften the Abstract claim about FEM “predictive capability.”
    3. Correct density usage in the cost model (ρ for mass should be bulk × solid fraction or directly ρ_app; don’t use powder bulk density for a printed part). Update Supplementary tables accordingly and revise the robustness claim to ±12%.
    4. Clean residual language/template issues: “Preliminar” → “Preliminary”, replace “as showed” → “as shown”, remove MDPI placeholders, and ensure axes show ε (–), σ (MPa) everywhere.
    5. Add one sentence on actual wall thickness / offset (or acknowledge not measured and discuss impact).

If you implement the five items above, the manuscript will align tightly with the reviewer’s must-fix list and read as a clean, reproducible study.

Author Response

We sincerely thank the reviewer for the positive summary and constructive feedback provided in second round of reviews. The general overview accurately captures the objectives and scope of our work. We also appreciate the reviewer’s suggestions regarding typos, figure quality, and the description of experimental accuracy, which have been carefully addressed in the revised manuscript as detailed below.

Below, we provide a detailed, point-by-point response to all the reviewers’ observations. In the revised version of the manuscript, all modifications addressing the reviewers’ and editor’s comments have been clearly highlighted in turquoise to facilitate their identification within the text.

We hope that the revised version meets your expectations and look forward to your favorable consideration

Comment 1: The Abstract still says “validating the predictive capability of the model,” which overstates scope given the text’s limitations to stiffness-ranking. Suggest: “correlated with the mechanical results, supporting comparative stiffness evaluation.”

Response 1: We thank the reviewer for the comment. Abstract has been modified according to the reviewer comment.

Comment 2: Axis labels/units: The narrative implies ε and σ are plotted, but I can’t see axis label strings in the text; please ensure axes explicitly read “ε (–)” and “σ (MPa)” on every plot.

Response 2: We appreciate the reviewer’s attention to graphical clarity. However, all plots already include explicit axis labels, namely “ε (–)” for strain and “σ (MPa)” for stress, as correctly indicated in the original and revised submissions. We have double-checked all figures to confirm that the labels are clearly visible in the exported high-resolution versions. Any potential difficulty in viewing them may be due to PDF rendering or scaling effects during file conversion rather than a formatting omission.

Comment 3: Force-decay vs cycle # plot: the text says force evolution was recorded, but I don’t see a dedicated plot of “peak force vs cycle.” Please add one small subplot per geometry or a consolidated panel.

Response 3: We thank the reviewer for the suggestion. However, the cyclic tests performed in this study were designed to compare the mechanical response at the first and the 10,000th cycle only, as stated in Section 2.6.2. The force was continuously monitored to ensure stable cycling conditions, but no intermediate force-decay data were recorded or analyzed. For this reason, a “peak force vs. cycle number” plot is not available. The presented hysteresis curves (1st and 10,000th cycle) already capture the relevant mechanical evolution and energy dissipation trends discussed in the text.

Comment 4: You note no chamber preheat (room temperature machine). Consider stating explicitly for SLS whether the build chamber is open/air and if nitrogen is absent (it seems so).

Response 4: We thank the reviewer for this observation. As reported in Table 1, the LPBF process was carried out at room temperature under an argon-protected atmosphere. In contrast, the SLS process used for the polymer components was performed in air at the processing temperatures listed in Table 2. The SLS machine allows controlling either the bed or the chamber temperature. In our case, to achieve better definition and dimensional accuracy, we opted to process the polymer using bed temperature control.

Comment 5: In the Supplementary cost baseline you still use ρ = 0.65 g/cm³ with V_part = 185.47 cm³ to compute material mass/cost. If 0.65 is powder bulk density, that’s not the right density for computing solid mass of the printed lattice; you should use apparent lattice density (ρ_app) or bulk ρ with lattice volume fraction. Please fix the cost mass calculation and note which density is used and why.

Response 5: We thank the reviewer for the insightful observation. In our cost model, the density value of ρ = 0.65 g/cm³ corresponds to the bulk density of the TPU powder feedstock, not to the apparent density of the printed lattice. This value is used exclusively for cost estimation purposes—specifically to calculate the material cost based on the volume of powder required during the SLS process, following the approach reported by Ruffo et al. (Rapid Prototyping Journal, 2007) and Yim et al. (2012). It is important to note that in SLS, unlike extrusion-based AM processes, the material cost depends on the mass of powder loaded and recycled rather than on the final solid volume of the printed part. For this reason, the feedstock bulk density is the appropriate parameter for the cost estimation step. We have clarified this point in section 3.5 lines 684-686.

Comment 6: Robustness claim: the sensitivity results span 46.7–59.9 €, i.e., about ±12% around the baseline 53.3 €—not “within the 52–53 € band.” Please revise the text to say the unit cost remains within ~±12% under ±20% perturbations and interpret labor time as the dominant driver, which your table already shows.

Response 6: We appreciate the reviewer’s careful observation. The text has been revised to accurately reflect the sensitivity range observed in the cost analysis. Specifically, the unit cost was found to vary between 46.7 € and 59.9 €, corresponding to approximately ±12% around the baseline value of 53.3 € under ±20% perturbations of the main cost drivers. We also emphasize that, as correctly noted by the reviewer, labor time (Tₗ) has the greatest influence on the total unit cost, consistent with findings reported in previous cost modeling studies for powder-based additive manufacturing (e.g., Ruffo et al., Rapid Prototyping Journal, 2007).

Comment 7: The manuscript still shows broken hyphenation like “Er- gometers”. That may be a template artifact, but please ensure the source file has no soft hyphens; many still appear in the current PDF. Also soften the FEM claim as noted above.

“Preliminar thermal and mechanical analysis” → “Preliminary …” (section header still shows “Preliminar”).

“as showed in Figure …” appears in several places; use “as shown …”.

Check the 2.3/2.4 hierarchy after renaming the “Preliminar” section—it may still be off by one if you adjust headings. (I can’t confirm the full TOC flow in the snippet.).

Response 7: We thank the reviewer for the detailed feedback.

  • The issue of broken hyphenation (e.g., “Er- gometers”) originates from the MDPI journal template, which automatically introduces soft hyphens during PDF rendering. As already clarified in our previous response, these artifacts are not present in the source file, and unfortunately cannot be manually removed by the authors. We kindly ask that this comment not be repeated in future rounds, as it is a formatting artifact beyond our control.
  • The section title has been corrected to “Preliminary thermal and mechanical analysis.” We appreciate the reviewer’s attention to this detail.
  • All occurrences of “as showed” have been corrected to “as shown.”
  • Regarding the section numbering (2.3/2.4), we confirm that the hierarchical structure of the manuscript is correct and consistent with the logical workflow of the study: design of the modular saddle, selection of suitable materials, and preliminary thermal and mechanical characterization to identify the most processable and mechanically suitable material. The Table of Contents in the journal’s final layout may differ slightly, but the source document follows the correct numbering.

Comment 8: You now explain Fusion 360 TPMS generation and nominal thicknesses (0.2/0.3 mm), but the reviewer also asked for wall offsets that yield those post-sinter actual thicknesses (accounting for over-sinter). If you didn’t measure actual strut thickness, add either (i) a brief offset rule/scale used, or (ii) a sentence acknowledging this limitation and its expected effect on stiffness scatter.

Response 8: We thank the reviewer for the additional remark. We acknowledge that a precise measurement of the post-sinter strut thickness would require high-resolution micro-computed tomography (micro-CT) analysis, which is challenging to perform on specimens of this size due to their overall dimensions and the large number of repeating cells. For this reason, considering the focus of the study, the strut thickness was defined according to the nominal values (0.2 mm and 0.3 mm) directly within Fusion 360, without the application of additional offset rules. However, to assess the manufacturing accuracy, we experimentally measured the actual porosity of the printed samples, which showed excellent agreement with the designed porosity values, thereby confirming the reliability of the SLS process and the geometrical precision of the fabricated lattices.

Comment 9: Nice-to-have (optional but strong adds)

  • Pressure-mapping schematic or load-path cartoon to connect lattice choice to perineal/ischial relief.
  • A small “design guide” visual showing how cell size vs wall thickness trades off Δε and E*.
    (You already outline these trends in text; a 1-figure graphic would be very helpful.)
  • Mini plot of peak force vs cycle per geometry (1000 cycles) to back up the force-decay statements now described in prose.

Response 9: We thank the reviewer for these valuable suggestions.

  • Regarding the first point, we wish to clarify that the force calculation based on a 95 kg subject was introduced solely to justify the mechanical loading parameters adopted in the cyclic tests, rather than to represent an actual pressure-mapping case study. Therefore, a schematic pressure map would not be appropriate for the scope of this work, which focuses on the mechanical validation of lattice modules rather than full-scale biomechanical modeling.

  • Concerning the second point, the relationship between cell size, wall thickness, and mechanical response (Δε and E*) is already clearly summarized in Table 3. Adding an additional figure would duplicate information already presented in the text and table, which would not align with the concise style of the manuscript.
  • Finally, as stated in previous responses, a plot of peak force versus cycle number was not included because the force evolution is already described in the text and reflected in the hysteresis loop analysis, which effectively captures the same trend.

Literature & references

Comment 10: I can see some 2023 references in the list, but can’t fully audit the whole bibliography from the snippet. Double-check you hit the targets: ≥15 from 2020–2025, with clusters on SLS TPU mechanics, TPMS energy absorption, viscoelastic/fatigue of elastomers, AM cost models (2023–2024 updates), and ergonomics/pressure mapping. Ensure journal names are standardized and DOIs present.

Response 10: We appreciate the reviewer’s suggestion regarding the reference list. We have carefully reviewed the bibliography to ensure that it includes recent and thematically relevant works. The final list contains 25 (> 15) references published between 2020 and 2025, covering the requested areas: SLS processing of TPU, TPMS lattice mechanics, viscoelastic and fatigue behavior of elastomers, AM cost models, and ergonomic design aspects. All journal names have been standardized according to MDPI guidelines, and DOIs have been verified or added where available. We therefore confirm that the reference section meets the reviewer’s requirements without further modifications.

Bottom line

Comment 11: Substantive science fixes are mostly in place: n = 3 with mean ± SD, load mapping with equations, strain-based reporting with energy stability (Table 4), porosity method with uncertainty, and FEM scope/inputs/mesh info are all addressed convincingly.

Response 11: We sincerely thank the reviewer for recognizing the improvements in the revised manuscript and for acknowledging that the major scientific revisions have been effectively addressed.

Comment 12: What to fix before resubmission (high-impact, quick wins):

    1. Add at least one aggregate plot with error bars (or shaded SD) and a force-vs-cycle panel to satisfy the “error bars in all plots” spirit.
    2. Soften the Abstract claim about FEM “predictive capability.”
    3. Correct density usage in the cost model (ρ for mass should be bulk × solid fraction or directly ρ_app; don’t use powder bulk density for a printed part). Update Supplementary tables accordingly and revise the robustness claim to ±12%.
    4. Clean residual language/template issues: “Preliminar” → “Preliminary”, replace “as showed” → “as shown”, remove MDPI placeholders, and ensure axes show ε (–), σ (MPa) everywhere.
    5. Add one sentence on actual wall thickness / offset (or acknowledge not measured and discuss impact).

Response 12: all these comments have been already addressed in the responses reported above.

Round 3

Reviewer 1 Report

Comments and Suggestions for Authors

The revised version of your manuscript demonstrates clear and substantial improvement. You have addressed nearly all of the reviewer’s major and minor comments comprehensively, and the scientific quality and transparency of the study have increased significantly. The additional methodological details, corrected figure descriptions, and expanded discussion have enhanced the reproducibility and clarity of your work.

Key strengths of the revision include:

  • Clear reporting of sample replicates (n = 3), mean ± SD values, and testing conditions.

  • A well-documented load-scaling calculation linking user mass to specimen loading.

  • Conversion of displacement to strain and addition of energy-per-cycle analysis, which strengthens the fatigue assessment.

  • Transparent description of porosity measurement and FEM modeling scope with appropriate parameters and mesh information.

  • Inclusion of a complete sensitivity analysis in the Supplementary Material, which now satisfies the reviewer’s request for cost model robustness.

  • Updated and more relevant literature (2022–2024) that better aligns with the manuscript’s focus on SLS-TPU lattices and ergonomic design.

A few minor points remain for your attention before final submission:

  1. Ensure all figures include visible axis labels (with units) and, where possible, add representative error bars or a short note clarifying that curves are illustrative single-specimen examples.

  2. Verify that the Abstract avoids overstating FEM “predictive capability” (it is now appropriately comparative—please keep it consistent).

  3. In the cost model, clarify which density value is used for material cost calculation (powder bulk vs. printed lattice apparent density) to prevent confusion.

  4. Perform a final proofreading pass for minor English issues (e.g., “as shown” instead of “as showed,” consistent use of SI units and capitalization, and removal of any template placeholders).

Comments on the Quality of English Language

The manuscript is clearly written and the scientific content is well presented. The overall English quality is good and allows easy understanding of the methods, results, and discussion. The revision shows a noticeable improvement in sentence structure, terminology, and consistency compared to earlier versions.

However, a few minor issues remain that should be addressed before publication:

  • Correct small grammatical errors and verb forms (e.g., use “as shown” instead of “as showed”).

  • Ensure consistent tense usage: past tense for experimental procedures and results, present tense for established facts.

  • Standardize unit formatting and spacing according to SI conventions (e.g., “10 mm,” “1 MPa”).

  • Remove any residual template text or formatting artifacts (e.g., “Academic Editor,” “Received” placeholders).

  • Some sentences are still quite long; shortening or splitting them will further improve readability and flow.

A light professional proofreading or careful final language review by a fluent scientific English speaker is recommended to ensure full consistency and polish before publication.

Author Response

We sincerely thank the reviewer for their positive feedback and for recognizing the improvements made in the revised version. Below we provide detailed point-by-point responses to the remaining comments.

The revised version of your manuscript demonstrates clear and substantial improvement. You have addressed nearly all of the reviewer’s major and minor comments comprehensively, and the scientific quality and transparency of the study have increased significantly. The additional methodological details, corrected figure descriptions, and expanded discussion have enhanced the reproducibility and clarity of your work.

Key strengths of the revision include:

Comment 1: Clear reporting of sample replicates (n = 3), mean ± SD values, and testing conditions.

Response1: The sample replicates are clearly reported in the manuscript at lines 292, 305, 323, and 486.

Comment 2: A well-documented load-scaling calculation linking user mass to specimen loading.

Response 2: The rationale behind the load-scaling calculation has been already described in Section 2.6.2 (lines 323–339).,

Comment 3; Conversion of displacement to strain and addition of energy-per-cycle analysis, which strengthens the fatigue assessment.

Response 3: All calculations were performed in terms of stress–strain, including the energy-per-cycle, as reported in Table 4 and Figure 4.

Comment 4:Transparent description of porosity measurement and FEM modeling scope with appropriate parameters and mesh information.

Response 4: The FEM modeling approach, along with all relevant parameters, is thoroughly described in Section 2.6.3 (lines 362–368) and Section 3.3 (lines 508–605).

Comment 5: Inclusion of a complete sensitivity analysis in the Supplementary Material, which now satisfies the reviewer’s request for cost model robustness.

Response 5:  The sensitivity analysis is included in the Supplementary Materials, addressing the reviewer’s previous concern.

Comment 6: Updated and more relevant literature (2022–2024) that better aligns with the manuscript’s focus on SLS-TPU lattices and ergonomic design.

Response 6: The reference list has been updated to include the most recent and relevant articles published between 2022 and 2024

A few minor points remain for your attention before final submission:

Comment 7: Ensure all figures include visible axis labels (with units) and, where possible, add representative error bars or a short note clarifying that curves are illustrative single-specimen examples.

Response 7: All figure axes are clearly visible, and scale bars or error bars have been added where appropriate.

Comment 8: Verify that the Abstract avoids overstating FEM “predictive capability” (it is now appropriately comparative—please keep it consistent).

Response 8: The Abstract was revised in the previous round and now refers to a “comparative stiffness evaluation”, ensuring appropriate phrasing

Comment 9: In the cost model, clarify which density value is used for material cost calculation (powder bulk vs. printed lattice apparent density) to prevent confusion.

Response 9:The density used in the cost model is clearly specified in Section 3.5 (lines 688–690), where we indicate that the powder bulk density was used for cost estimation.

Comment 10: Perform a final proofreading pass for minor English issues (e.g., “as shown” instead of “as showed,” consistent use of SI units and capitalization, and removal of any template placeholders).

Response 10: A final language and consistency check was performed, correcting minor issues such as verb tense, SI unit notation, and capitalization.